# Combined ADAMTS10 and ADAMTS17 inactivation exacerbates bone shortening and skin phenotypes

Nandaraj Taye[1,*], Stylianos Z Karoulias[1,*] , Zerina Balic[1], Lauren W Wang[3,4], Belinda B Willard[2], Daniel Martin[3] , Daniel Richard[5], Alexander S Okamoto[5] , Terence D Capellini[5,6], Suneel S Apte[3,4], Dirk Hubmacher[1,7]

Weill–Marchesani syndrome (WMS) is characterized by severe short stature, joint contractures, tight skin, heart valve and eye anomalies. WMS is caused by biallelic mutations in *ADAMTS10, ADAMTS17,* or *LTBP2,* or mono-allelic mutations in *FBN1.* Because bone growth is driven by chondrocyte proliferation and hypertrophy in the growth plates, the genetics of WMS suggests that the affected extracellular matrix (ECM) proteins contribute to chondrocyte and growth plate function. Here, we show that *Adamts10;Adamts17* double knockout (DKO) mice have significant postnatal lethality and exacerbated bone shortening, which correlated with a narrower hypertrophic zone in their growth plates. Potential ADAMTS17 substrates identified by N-terminomics and yeast-2-hybrid screening revealed the ECM proteins fibronectin (*FN1*) and collagen VI (*COL6A2*). In primary ADAMTS10- and ADAMTS17-deficient skin fibroblasts, fibronectin deposition was impaired concomitant with aberrant intracellular accumulation of fibrillin-1. These findings support a role for ADAMTS17 in ECM protein secretion and assembly. Mechanistically, ADAMTS17 appears to be a critical regulator of ECM protein secretion or pericellular matrix assembly, whereas ADAMTS10 likely modulates ECM formation at later stages.

## Introduction

Acromelic dysplasias are a group of genetic conditions resulting in severe short stature and shortening of distal limb elements (Le Goff & Cormier-Daire, 2009; Stanley et al, 2020). Among them, Weill–Marchesani syndrome (WMS) has been characterized genetically resulting from mutations in distinct, but functionally related genes, encoding ECM proteins (Stanley et al, 2020; Marzin et al, 2023).

Biallelic pathogenic variants in *ADAMTS10* cause WMS1 (Dagoneau et al, 2004; Kutz et al, 2008; Morales et al, 2009). Distinct WMS subtypes can also be caused by mono-allelic mutations in fibrillin-1 (*FBN1,* WMS2) or biallelic mutations in *LTBP2* (WMS3) or *ADAMTS17* (WMS4), suggesting that these four genes may act together in pathways that regulate development and homeostasis of affected tissues (Faivre et al, 2003b; Morales et al, 2009; Haji-Seyed-Javadi et al, 2012; Cecchi et al, 2013; Karoulias et al, 2020a; Evans et al, 2020). WMS is characterized by short stature, lens dislocation, microspherophakia and other eye anomalies, progressive joint stiffness, and tight skin (Faivre et al, 2003a; Marzin et al, 2023). Lens dislocation and/or changes in the outflow trackt can block the drainage of aqueous humor from the anterior chamber of the eye and cause glaucoma in WMS. WMS1 also has cardiovascular manifestations such as patent ductus arteriosus, pulmonary valve dysplasia, which can lead to pulmonary stenosis, and in rare cases aortic aneurysms (Faivre et al, 2003a; Dagoneau et al, 2004; Cecchi et al, 2013). In dogs, mutations in *ADAMTS10* and *ADAMTS17* were associated with primary open angle glaucoma (Farias et al, 2010; Gould et al, 2011; Kuchtey et al, 2011, 2013; Oliver et al, 2015). Homozygosity of a glaucoma-causing canine *ADAMTS17* variant was also associated with short stature in several dog breeds (Jeanes et al, 2019). WMS-causing ADAMTS10 and ADAMTS17 mutations are distributed over the entire molecule and result in loss-of-function or haploinsufficiency due to impaired secretion (Karoulias et al, 2020a; Evans et al, 2020; Stanley et al, 2020; Marzin et al, 2023). The fact that mutations in *ADAMTS10* and *ADAMTS17* each cause WMS suggests that these genes cooperate, have superimposed mechanisms, or act in the same pathways that regulate the development or homeostasis of affected tissues, including the growth plate, which drives bone growth, the skin, and the eye (Hubmacher & Apte, 2011; Karoulias et al, 2020b). ADAMTS10 and ADAMTS17 each bind to fibrillin-1, which may provide a scaffold for their tissue-specific deposition or functional regulation (Kutz et al, 2011; Hubmacher et al, 2017). In addition,

---

[1]Orthopedic Research Laboratories, Leni and Peter W. May Department of Orthopedics, Icahn School of Medicine at Mount Sinai, New York, NY, USA [2]Proteomics and Metabolomics Core, Cleveland Clinic Lerner Research Institute, Cleveland, OH, USA [3]Department of Biomedical Engineering, Cleveland Clinic Lerner Research Institute, Cleveland, OH, USA [4]Department of Orthopaedic Surgery, Cleveland Clinic Orthopaedic and Rheumatologic Institute, Cleveland, OH, USA [5]Human Evolutionary Biology, Harvard University, Cambridge, MA, USA [6]Broad Institute of MIT and Harvard, Cambridge, MA, USA [7]Mindich Child Health and Development Institute, Icahn School of Medicine at Mount Sinai, New York, NY, USA

Correspondence: dirk.hubmacher@mssm.edu
Stylianos Z Karoulias's present address is Regeneron Pharmaceuticals, Tarrytown, NY, USA
Zerina Balic's present address is Weill Cornell BCMB Program, Sloan Kettering Institute, New York, NY, USA
*Nandaraj Taye and Stylianos Z Karoulias contributed equally to this work

---

ADAMTS10 was shown to promote the assembly of fibrillin-1 in cell culture (Kutz et al, 2011).

ADAMTS proteases are involved in diverse biological processes including tissue morphogenesis and homeostasis (Apte, 2020; Satz-Jacobowitz & Hubmacher, 2021). Mutations in several ADAMTS proteases cause birth defects and inherited connective tissue disorders in humans and other species (Mead & Apte, 2018). In addition, ADAMTS proteases contribute to the progression of acquired disease, such as the aggrecanase ADAMTS5 in osteoarthritis (Santamaria, 2020). These diverse roles are attributed to the actions of ADAMTS proteases on distinct substrates. Recognized ECM substrates for ADAMTS proteases include proteoglycans such as aggrecan and versican, the ECM scaffolding proteins fibrillin-1, fibrillin-2, and fibronectin, and several collagens (Satz-Jacobowitz & Hubmacher, 2021). Most ADAMTS proteases are secreted as inactive zymogens whose activation requires proteolytic removal of their prodomains by furin or other proprotein convertases (Wang et al, 2004; Koo et al, 2006, 2007; Longpre et al, 2009). ADAMTS10, however, lacks a canonical furin recognition site and is poorly processed by furin. Consistent with poor furin processing, ADAMTS10 appears to be an inefficient protease, although, once activated by mutagenesis to restore a canonical furin-processing site, ADAMTS10 could cleave fibrillin-1 and fibrillin-2 in vitro (Kutz et al, 2011; Wang et al, 2019). However, alternative furin-independent mechanisms of ADAMTS10 activation in vitro or in vivo remain elusive. ADAMTS17 is secreted as an active protease, but fragments itself extensively and efficiently by autocatalysis, including within the catalytic domain, before its release from the surface of HEK293 cells (Hubmacher et al, 2017). These findings suggested that ADAMTS17 may act as a protease in the secretory pathway or at the cell surface. ADAMTS17 substrates other than itself have not been identified. We showed previously that ADAMTS17 interacted with fibrillin-1 and fibrillin-2 but did not cleave either (Hubmacher et al, 2017; Balic et al, 2021).

*Adamts10* and *Adamts17* inactivation in mice was previously reported with differential phenotypes. Whereas *Adamts10* knockout (KO) mice did not display short stature or bone shortening, *Adamts17* KO mice had shorter bones because of growth plate abnormalities (Oichi et al, 2019; Wang et al, 2019). Interestingly, the knock-in of a WMS mutation into the mouse *Adamts10* locus resulted in short stature and growth plate abnormalities (Mularczyk et al, 2018). In addition, knock-in of an *ADAMTS10* mutation that causes glaucoma in dogs into the mouse *Adamts10* locus also resulted in short stature (Wu et al, 2021). Together, these findings support a role for ADAMTS10 and ADAMTS17 in regulating bone growth and thus height. If and how ADAMTS10 and ADAMTS17 cooperate in regulating bone growth and in the formation and maintenance of other tissues affected in WMS is not known. Here, we investigated the genetic interactions of ADAMTS10 and ADAMTS17 by analyzing the bone and skin phenotypes of *Adamts10;Adamts17* double KO (DKO) mice. For insights on molecular mechanisms, we evaluated potential ADAMTS17 substrates and binding partners identified by N-terminomics and yeast-2-hybrid screening, respectively. The findings of our studies, taken together with prior work, strongly suggest a cooperative role for these proteases in skeletal growth and provide a putative molecular basis for their ECM-regulatory activities.

# Results

## Combined *Adamts10* and *Adamts17* inactivation resulted in early postnatal mortality and reduced body size

To generate *Adamts10;Adamts17* DKO mice, we combined a previously published *Adamts10* KO allele with a novel *Adamts17* KO allele. The *Adamts10* KO allele was generated by replacing 41 bp of *Adamts10* exon 5 with an IRES-*lacZ-neo* cassette (Wang et al, 2019). The *Adamts17* KO allele was generated by CRISPR/Cas9-induced nonhomologous end joining with guide RNAs targeting *Adamts17* exon 3 (Fig 1A). The resulting dinucleotide insertion (AT) in exon 3 of *Adamts17* (*Adamts17* 670_671insAT, NM_001033877.4) caused a reading frame shift (p.I190fsX12) with a premature termination codon that is expected to trigger nonsense-mediated mRNA decay (Fig 1A). The AT insertion in *Adamts17* was verified by Sanger sequencing of a PCR product amplified from template DNA isolated from toe tissue with *Adamts17*-specific primers flanking exon 3 (Fig 1B). Mice homozygous for the *Adamts17* AT insertion are referred to as *Adamts17* KO. To confirm that the AT insertion resulted in a reduction of ADAMTS17 mRNA and protein levels, we first quantified ADAMTS17 mRNA levels isolated from WT and *Adamts17* KO lung using quantitative real-time PCR. Strongly reduced ADAMTS17 mRNA levels in lung tissue from *Adamts17* KO mice indicated that the AT-induced reading frameshift resulted in significant ADAMTS17 mRNA degradation (Fig 1C). In addition, the loss of ADAMTS17 protein was validated by immunostaining in tissues and primary skin fibroblasts isolated from WT and *Adamts17* KO and *Adamts10;Adamts17* DKO mice using a commercial monoclonal ADAMTS17 antibody, which we have characterized previously (Hubmacher et al, 2017). In sections through WT skin and the tibial growth plate, ADAMTS17 immunoreactivity was apparent around hair follicles and hypertrophic chondrocytes, respectively. This signal was absent or strongly reduced in comparable sections from *Adamts17* KO or *Adamts10;Adamts17* DKO mice (Fig 1D). In primary WT skin fibroblasts, the ADAMTS17 signal was most intense in perinuclear regions, where the endoplasmic reticulum and Golgi are located, and in patches between cells (Fig 1D, right). Like in tissue sections, the ADAMTS17 signal was absent in DKO fibroblasts. Thus, genetic inactivation of ADAMTS17 eliminated ADAMTS17 mRNA and protein in relevant tissues and primary cells, which at the same time validated the specificity of the monoclonal ADAMTS17 antibody.

*Adamts17* KO mice were born at the expected Mendelian ratio and were viable (Fig 1E). To generate DKO mice, we first crossbred *Adamts10* Het and *Adamts17* Het mice to generate *Adamts10; Adamts17* double-heterozygous mice (Fig 1F). *Adamts10* Het;*Adamts17* Het mice were then intercrossed to generate offspring with all allelic combinations, among which WT, *Adamts10* KO; *Adamts17* KO, and *Adamts10;Adamts17* DKO mice were the focus of subsequent analyses. In principle, this breeding scheme allows comparison of the phenotypes from littermates. However, because of the low predicted percentages for WT and DKO mice (6.25%) per litter in these crosses, we also intercrossed *Adamts10* Het or *Adamts17* Het mice to generate the respective WT and individual KOs as age- and sex-matched controls for DKO mice. We first analyzed the *Adamts10* and *Adamts17* genotype distribution of 180 mice at

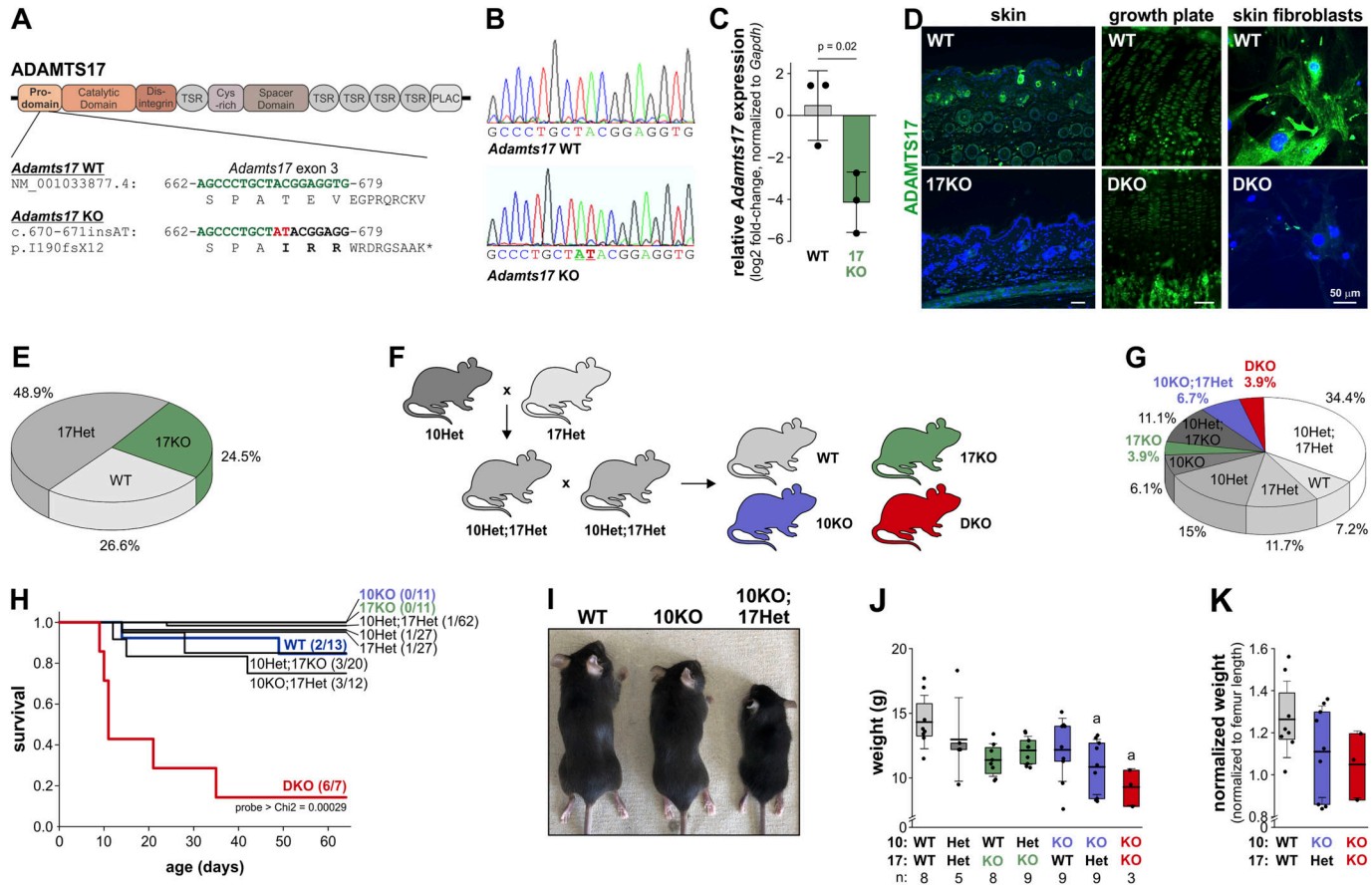

**Figure 1. Generation of *Adamts17* KO and *Adamts10;Adamts17* DKO mice.**
**(A)** Domain organization of ADAMTS17 shows location and targeting of exon 3 by CRISPR/Cas9 gRNA to induce nonhomologous end joining. The nucleotide and amino acid sequence of the ADAMTS17 WT allele (green) and after AT insertion (red) are indicated. The dinucleotide insertion induced a frameshift, which resulted in a premature stop codon after 12 amino acids. **(B)** Sanger sequencing traces of a PCR product generated with primers flanking exon 3 showing the AT insertion (underlined) in the *Adamts17* KO. **(C)** Quantitative real-time PCR using cDNA prepared from WT and *Adamts17* KO lung tissue as a template shows significant reduction of ADAMTS17 mRNA in the KO (n = 3). **(D)** Micrographs of ADAMTS17 immunostaining of sections through WT and *Adamts17* KO skin (left), DKO growth plates (middle), and of primary DKO mouse skin fibroblasts (right). The signal in the dermis around hair follicles, in growth plate chondrocytes, and in fibroblasts and their ECM originating from the monoclonal ADAMTS17 antibody was strongly reduced in KO and DKO tissues and cells, indicating lack of ADAMTS17 protein in *Adamts17* KO mice. **(E)** Pie chart showing Mendelian distribution of genotypes recovered from *Adamts17* Het intercrosses at the time of genotyping (P7–P10) (n = 94 mice). **(F)** Breeding scheme to generate WT, *Adamts10* KO (10KO), *Adamts17* KO (17KO), and DKO mice. **(G)** Pie chart showing distribution of genotypes recovered from *Adamts10* Het;*Adamts17* Het intercrosses at P7–P10 (n = 180 mice). Statistical analysis was performed using Chi square calculation. **(H)** Kaplan–Meier survival analysis of DKO mice. The numbers of observed dead/total mice for the individual genotypes are indicated in brackets. Statistical significance was determined using a log-rank test. **(I)** Whole mount images of WT, 10KO, 10KO;17Het mice at 4 wk of age show progressive reduction in body size. **(J)** Bar graphs showing body weights of 4-wk-old mice of the indicated genotypes. The number of mice is indicated below the genotypes. **(I, K)** Bar graphs showing body weight normalized to average femur length for the genotypes that were significantly different in (I). Bars in (C) indicate mean values and whiskers the SD. In (J, K) floating bars indicate the 25th–75th percentile range, lines the mean value, and whiskers the SD. **(C, J, K)** Statistical differences in (C) were determined using a two-sided *t* test and (J, K) were using a one-way ANOVA with post hoc Tukey test. a, $P < 0.05$ compared with WT.

postnatal day (P) 7-P10 and determined statistically significant deviations from the expected Mendelian ratios using $\chi^2$ calculation (Fig 1G). *Adamts17* KO (3.9% versus 6.25% expected), *Adamts10* KO;*Adamts17* Het (6.7% versus 12.5% expected), and DKO (3.9% versus. 6.25% expected) mice were present in significantly lower numbers than expected and the percentage of *Adamts10;Adamts17* double-heterozygous mice was significantly higher (34.4% versus. 25% expected). This resulted in a $\chi^2$ value of 18.09 (8 degrees of freedom) and a *P*-value of <0.05, suggesting reduced postnatal viability or embryonic lethality due to reduced *Adamts17* gene dosage or the combined absence of *Adamts10* and *Adamts17*. Because we observed early postnatal lethality of *Adamts10;Adamts17* DKO mice,

we quantified postnatal survival with Kaplan–Meier survival analysis, where we confirmed significant postnatal mortality of DKO mice with 50% survival at 23 d (±7.1 d) after birth (probe > $\chi^2$ = 0.00029, log-rank test) (Fig 1H). The cause of death is unknown. In addition to reduced survival, we noted reduced body size of DKO mice, which correlated with several genotypes, most notably *Adamts17* KO mice, *Adamts10* KO;*Adamts17* Het mice, and DKO mice (Fig 1I). These size differences were also apparent from body weight measurements at 4 wk of age where the weights of *Adamts10* KO;*Adamts17* Het and DKO mice were significantly reduced compared with WT (Fig 1J). However, after normalization of the body weight to the average femur lengths, these differences

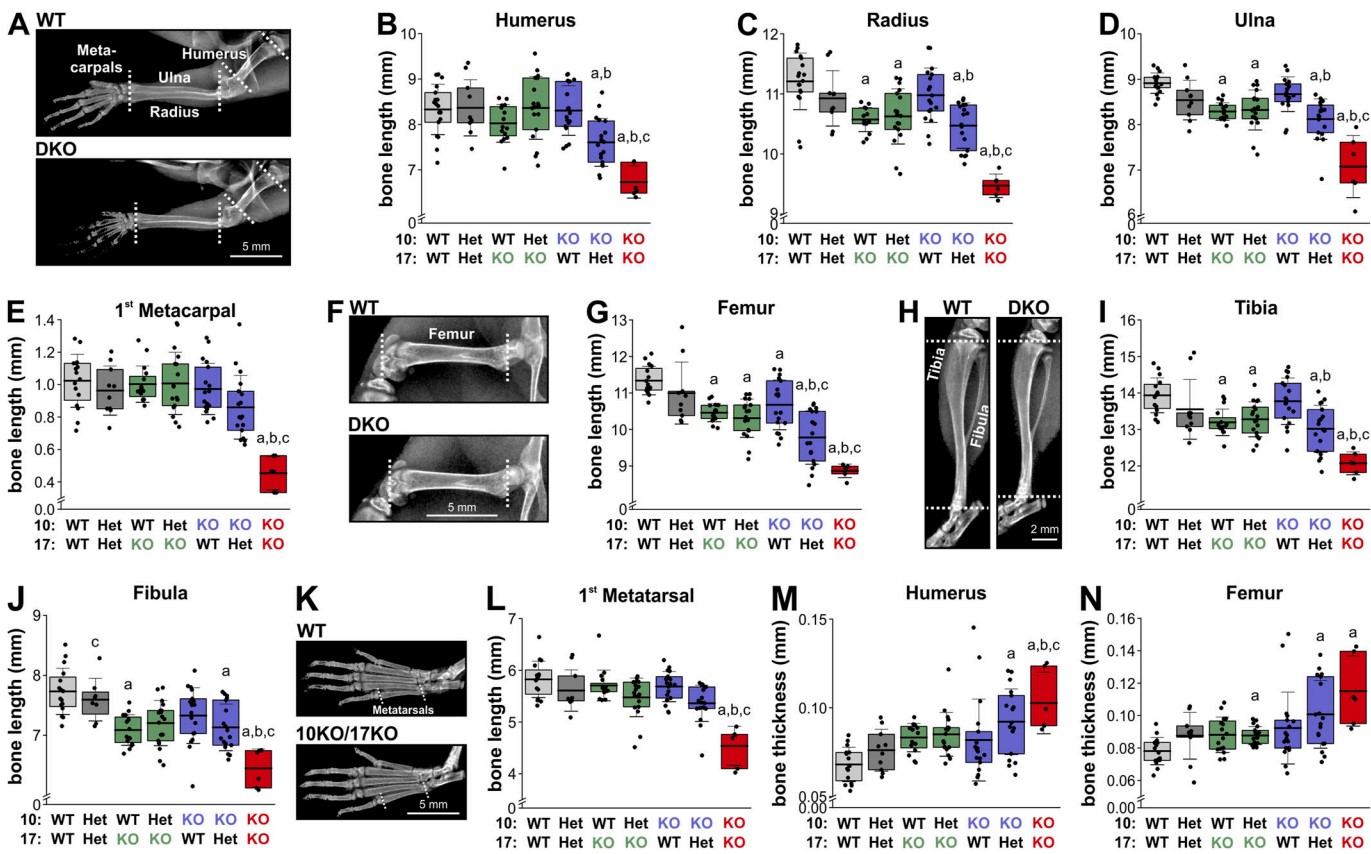

**Figure 2. Exacerbated bone shortening in Adamts10;Adamts17 DKO mice.**
**(A)** X-ray images showing WT and DKO forelimbs. **(B, C, D, E)** Bar graphs showing lengths of humerus (B), radius (C), ulna (D), and first metacarpal (E) for all genotypes. **(F)** X-ray images showing WT and DKO femur. **(G)** Bar graphs showing femoral length for all genotypes. **(H)** X-ray images showing WT and DKO tibia and fibula. **(I, J)** Bar graphs showing lengths of tibia (I) and fibula (J) for all genotypes. **(K)** X-ray images showing WT and DKO hind paw bones. **(L)** Bar graphs showing length of first metatarsal for all genotypes. **(M, N)** Bar graphs showing thickness of humerus (M) and femur (N) at mid-shaft. For number of mice, see Fig 1I. All mice were 4 wk old at the time of X-ray imaging. Bones from both limbs were measured, and measurements from male and female mice were combined. In (B, C, D, E, G, I, J, L, M, N), floating bars indicate the 25th–75th percentile range, lines the mean value, and whiskers the SD. Statistical differences in (B, C, D, E, G, I, J, L, M, N) were determined using a one-way ANOVA with post hoc Tukey test. a, $P < 0.05$ compared with WT; b, $P < 0.05$ compared with *Adamts10* KO; c, $P < 0.05$ compared with *Adamts17* KO.

became nonsignificant, suggesting proportionate short stature (Fig 1K).

## Combined *Adamts10* and *Adamts17* depletion exacerbated bone shortening

To determine if *Adamts10* and *Adamts17* gene dosage affected bone length, we quantified the lengths of forelimb and hind limb bones from X-ray images taken at 4 wk of age (Fig 2A–L). Overall, *Adamts10;Adamts17* DKO mice had the shortest bones across all genotypes when compared with WT or individual KOs. In addition, *Adamts17* KO bones were significantly shorter than WT, except for the humerus, the first metacarpal, and the first metatarsal. Notably, deletion of one *Adamts17* allele in *Adamts10* KO mice exacerbated bone shortening, but not vice versa, except in the fibula, the first metacarpal, and the first metatarsal. The lengths of the first metacarpal and first metatarsal were significantly shorter in DKO mice than in WT, *Adamts10* KO, and *Adamts17* KO, but the same bones were not shorter in the individual KOs than in WT (Fig 2E and L). The lengths of *Adamts10;Adamts17* double-

heterozygous bones were not significantly shorter than WT bones. Because we previously observed increased bone width in another acromelic dysplasia model (geleophysic dysplasia due to *Adamtsl2* deficiency), we measured the width of humeri and femora (Hubmacher et al, 2019). The mid-shaft width in DKO mice was significantly greater than WT, *Adamts10* KO, or *Adamts17* KO (humerus), or than WT (femur) (Fig 2M and N). In addition, the widths of the *Adamts10* KO;*Adamts17* Het humerus and femur were increased compared with WT, but not the individual KOs. Collectively, combined inactivation of ADAMTS10 and ADAMST17 exacerbated bone shortening compared with the individual KOs with *Adamts17* having an apparently stronger gene dosage effect over *Adamts10*.

## Combined *Adamts10* and *Adamts17* deficiency affects growth plate architecture and differentially compromises chondrocyte function

Because bone growth is largely driven by growth plate activity, i.e., the proliferation and hypertrophic expansion of growth plate

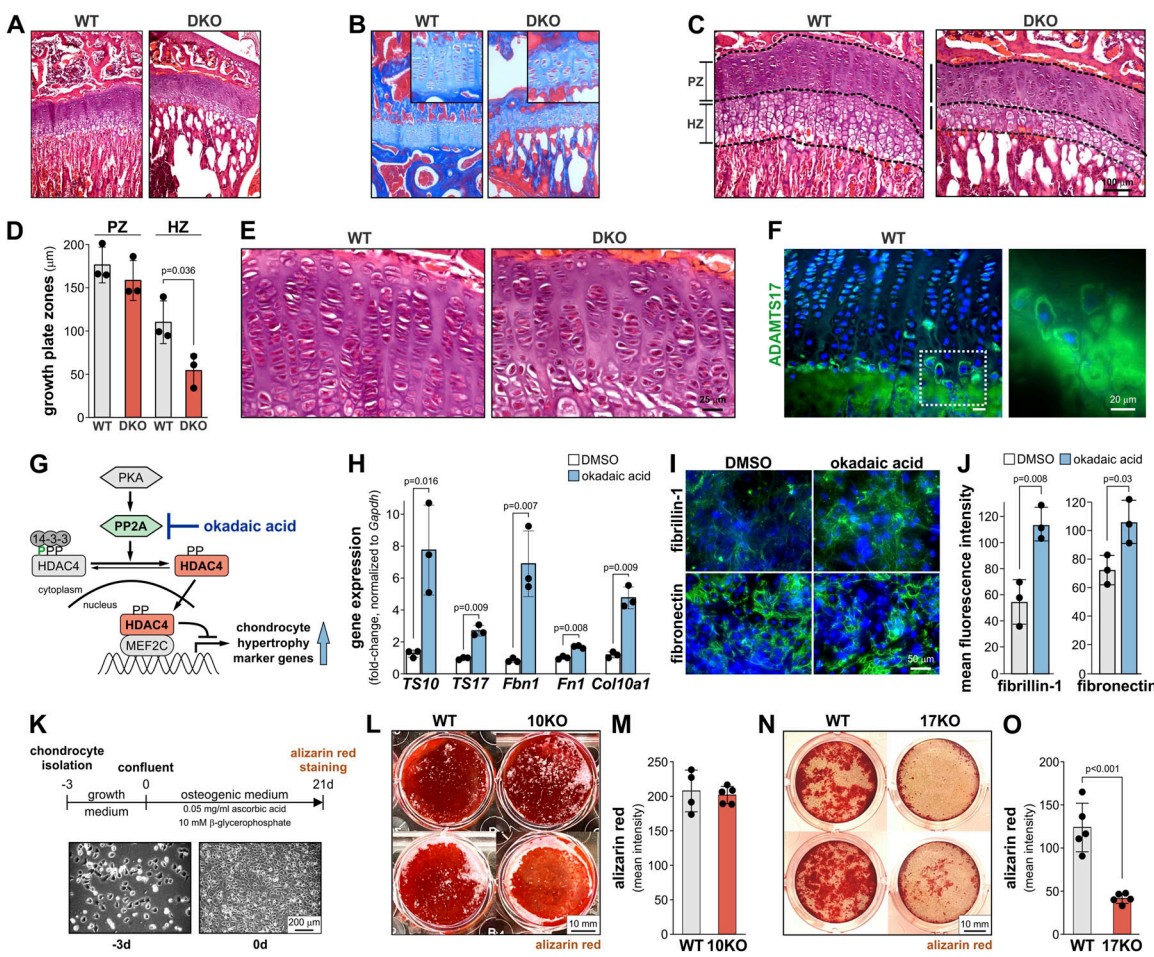

**Figure 3. ADAMTS10 and ADAMTS17 regulate growth plate function and chondrocyte hypertrophy.**
**(A, B)** Low magnification images of hematoxylin and eosin- (A) or Masson's trichrome–stained (B) sections through growth plates of 4-wk-old WT and *Adamts10*;
*Adamts17* DKO mice showing overall growth plate architecture and collagen content. Inset in (B) shows higher magnification of growth plate. **(C)** Higher magnification
images of sections through growth plates of 4-wk-old WT and *Adamts10;Adamts17* DKO mice. Proliferative (PZ) and hypertrophic (HZ) zones are outlined with dashed
lines. **(D)** Bar graphs showing widths of PZ and HZ from WT and DKO growth plates. Data points represent the average of multiple measurements across the growth
plate zone from n = 3 mice. **(A, E)** Higher magnification of growth plate from images in (A) showing disorganized proliferative zone in DKO growth plates. **(F)** Micrograph of
ADAMTS17 immunostaining of hypertrophic chondrocytes at the cartilage-bone interface. The boxed area is magnified in the right-hand panel. **(G)** Schematic
representation of the mechanism of action of okadaic acid in de-repressing chondrocyte hypertrophy genes. **(H)** Bar graphs showing relative changes of ADAMTS10
(TS10), ADAMTS17 (TS17), FBN1, FN1, and COL10A1 mRNA levels 24 h after treatment of primary chondrocytes with 50 nM okadaic acid or DMSO (n = 3 replicates).
**(I)** Micrographs of immunostaining of fibrillin-1 (FBN1) and fibronectin (FN) deposition in the ECM of primary chondrocytes 3 d after treatment with okadaic acid or DMSO
only. **(I, J)** Quantification of mean fluorescence intensity from (I) (n = 3 fields of view). **(K)** Schematic representation of osteogenic differentiation of P5 primary rib
chondrocytes isolated from *Adamts10* KO or *Adamts17* KO mice. The bottom panels show bright-field micrographs of freshly isolated primary chondrocytes (−3 d, left)
and confluent chondrocytes (0 d, right). **(L)** Micrographs of two individual wells/genotype of primary WT or *Adamts10* KO (10KO) chondrocytes stained with alizarin red
after 21 d of culture in osteogenic medium. **(M)** Bar graph showing quantification of mean signal intensity of alizarin red deposits (isolates from n = 4–5 biological
replicates/genotype). **(N)** Micrographs of two individual wells/genotype of primary WT or *Adamts17* KO chondrocytes stained with alizarin red after 21 d of culture in
osteogenic medium. **(O)** Bar graph showing quantification of mean signal intensity of alizarin red deposits (isolates from n = 5 biological replicates/genotype). In (D, H, J,
M, O), bars indicate mean values and whiskers the SD. **(F)** Statistical significance in (D, H, J, M, O) was calculated with a two-sided *t* test and in (F) with one-way ANOVA
followed by post hoc Tukey test.

chondrocytes, we analyzed growth plate architecture and characterized primary chondrocyte behavior in the absence of ADAMTS10 and ADAMTS17 (Kember, 1972; Breur et al, 1991). Compared with WT growth plates, DKO growth plates showed a narrower hypertrophic zone, whereas the proliferative zone was unchanged (Fig 3A–E). Masson's trichrome staining confirmed the overall disorganization of the DKO growth plate (Fig 3B). Whereas the collagen content in the growth plate was similar, increased collagen intensity was observed in the adjacent bone elements. Apparently empty spaces

below the growth plate in the DKO images were attributed to an artifact during tissue and section preparation resulting in the loss of bone marrow cells. In addition, the columnar organization of proliferating chondrocytes appeared to be irregular with more spacing between chondrocyte columns (Fig 3E). By immunostaining, we localized ADAMTS17 in the pericellular matrix of hypertrophic chondrocytes in WT growth plates, close to the cartilage-bone interface (Fig 3F). This signal was strongly reduced in the *Adamts17* KO growth plate (see Fig 1D).

To further probe if *Adamts10* and *Adamts17* expression was regulated during chondrocyte hypertrophy, we treated primary WT rib chondrocytes with okadaic acid, a protein phosphatase 2A inhibitor, which results in de-repression of chondrocyte hypertrophy genes (Fig 3G) (Kozhemyakina et al, 2009; Dy et al, 2012). qRT-PCR showed significant up-regulation of *Adamts10* and *Adamts17* mRNA levels 24 h after treatment of primary WT chondrocytes with okadaic acid compared with DMSO-treated control chondrocytes (Fig 3H). Increased COL10A1 mRNA levels, a marker for hypertrophic chondrocytes, served as a positive control for okadaic acid–mediated de-repression of chondrocyte hypertrophy genes. In addition to ADAMTS10 and ADAMTS17, mRNAs for genes encoding the ECM proteins fibrillin-1 (*Fbn1*) and fibronectin (*Fn1*) were also induced. Fibrillin-1 binds to ADAMTS10 and ADAMTS17 and mutations in *FBN1* cause WMS2 (Faivre et al, 2003b; Kutz et al, 2011; Hubmacher et al, 2017). Fibronectin forms the ECM scaffold required for fibrillin-1 deposition in the ECM of mesenchymal cells (Sabatier et al, 2009). Using immunostaining, we confirmed increased fibrillin-1 and fibronectin ECM deposition in primary WT chondrocytes after okadaic acid treatment (Fig 3I and J).

Finally, we investigated the implications of ADAMTS10 and ADAMTS17 deficiency on primary rib chondrocyte hypertrophy and their capacity to deposit calcium as hydroxyapatite in their ECM, which is a characteristic of terminal hypertrophic chondrocytes (Fig 3K). Chondrocytes isolated from *Adamts10* KO ribs showed no difference in alizarin red-positive calcium mineral deposition after 21 d, whereas calcium mineral deposition by ADAMTS17-deficient primary chondrocytes was significantly reduced, suggesting differential roles or differential compensation for ADAMTS10 and ADAMTS17 in chondrocyte differentiation or mineral deposition (Fig 3L–O).

To gain further insights into the epigenetic and transcriptomic regulation of *ADAMTS10* and *ADAMTS17* during human embryonic growth plate development, we data-mined recent assays for transposase-accessible chromatin with sequencing (ATAC-seq) and RNA-transcriptomic data sets, generated from microdissected cartilaginous human fetal appendicular skeletal elements (Fig 4A) (Richard et al, 2024; Okamoto et al, 2025 *Preprint*). We first identified open chromatin regions by ATAC-seq within ±100 kb of *ADAMTS10* and *ADAMTS17* (Fig 4B and C); each interval containing regulatory elements, such as enhancers, promoters, and repressor sequences, likely drives expression of the nearby gene. For *ADAMTS10*, out of 10 total elements within 100 kbp, we identified three cartilage open chromatin regions that were shared by most or all autopod elements (phalanges, metatarsals, and metacarpals), but were absent in stylopod or zeugopod elements, i.e., proximal and distal ends of all major long bones (Fig 4B, Table 1). For *ADAMTS17*, we observed a larger number of regulatory elements than for *ADAMTS10*, consistent with the larger size of *ADAMTS17*. At this locus, 23 cartilage open chromatin regions were identified within or in close vicinity to the *ADAMTS17* gene body (Fig 4C, Table 2). Whereas three *ADAMTS17* open chromatin regions were identified as accessible in all autopod elements and eight in most, two were specific to individual skeletal elements of the hind limb autopod (metatarsal V). We next analyzed *ADAMTS10* and *ADAMTS17* gene expression in human embryonic skeletal elements by comparing normalized read counts as a measure of ADAMTS10 and

ADAMTS17 mRNA abundance (Fig 4D and E). In stylopodial and zeugopodial elements, normalized read counts for *ADAMTS10* were generally higher than for *ADAMTS17*, suggesting increased expression (Fig 4D). We did not observe a distinct pattern of *ADAMTS10* or *ADAMTS17* expression based on specific skeletal elements, suggesting that both genes were expressed in autopods of the hind limb and forelimb. Finally, we identified several predicted binding sites of key chondrogenic and osteogenic transcription factors, including SOX9, RUNX2, and ATF4, within 5 kb upstream of the transcriptional start site of *ADAMTS10* and *ADAMTS17* using the Search Motif Tool in the Eukaryotic Promoter Database (Fig 4F and G) (Dreos et al, 2017).

Collectively, these data suggest specific regions of chromatin accessibility in the vicinity of *ADAMTS10* and *ADAMTS17* and *ADAMTS10* and *ADAMTS17* gene expression during human embryonic cartilage development. In addition, the mapping of chondrogenic and osteogenic transcription factor binding sites in the *ADAMTS10* and *ADAMTS17* promoter regions support the regulation of *Adamts10* and *Adamts17* expression during growth plate development.

### *Adamts10* and *Adamts17* inactivation compromised skin development and ECM deposition by skin fibroblasts

*Adamts10;Adamts17* DKO skin easily detached and ripped during shaving. We further investigated this phenomenon by measuring the thickness of the dermal sub-layers in Masson's trichrome stained cross sections (Fig 5A–E). The overall thickness of *Adamts10* KO and *Adamts17* KO dorsal skin was reduced compared with WT skin, which was further exacerbated in DKO skin (Fig 5B). Epidermal layer thickness was slightly but significantly reduced in DKO skin compared with WT and *Adamts10* KO skin but not compared with *Adamts17* KO skin (Fig 5C, left). The thickness of *Adamts10* KO and *Adamts17* KO dermis and hypodermis was significantly reduced compared with WT and further reduced in the DKO (Fig 5C, middle, right). The panniculus carnosus (p. carnosus) muscle, which underlies mouse skin but not human skin, was similarly significantly thinner in *Adamts17* KO and DKO skin sections compared with WT and *Adamts10* KO (Fig 5D). The proportion of the hypodermis to overall skin thickness was greatly reduced in DKO skin, whereas the proportions of the dermis and p. carnosus were both increased (Fig 5E). A similar reduction in the hypodermal layer and increase in the dermal layer were evident in *Adamts17* KO skin and to a lesser extent in *Adamts10* KO skin. We also observed a significant reduction in the number of hair follicles in DKO skin compared with all other genotypes, but there were no significant changes in the *Adamts10* KO or *Adamts17* KO when compared with WT skin (Fig 5F).

To identify potential cellular origins of ADAMTS10 and ADAMTS17 in the skin, we data-mined the Hair-GEL database, which contains single-cell transcriptomic data from E14.5 and P5 skin with a focus on hair follicle development (Sennett et al, 2015; Rezza et al, 2016). At E14.5, differential expression of *Adamts10* and *Adamts17* was noted in the dermal condensate, where *Adamts10* expression was high, and in the placode and epidermis, where *Adamts17* expression was high (Fig 5G). In fibroblasts, Schwann cells, or melanocytes, *Adamts10* and *Adamts17* were expressed at similar levels. At P5, *Adamts10* was strongly expressed in subtypes of

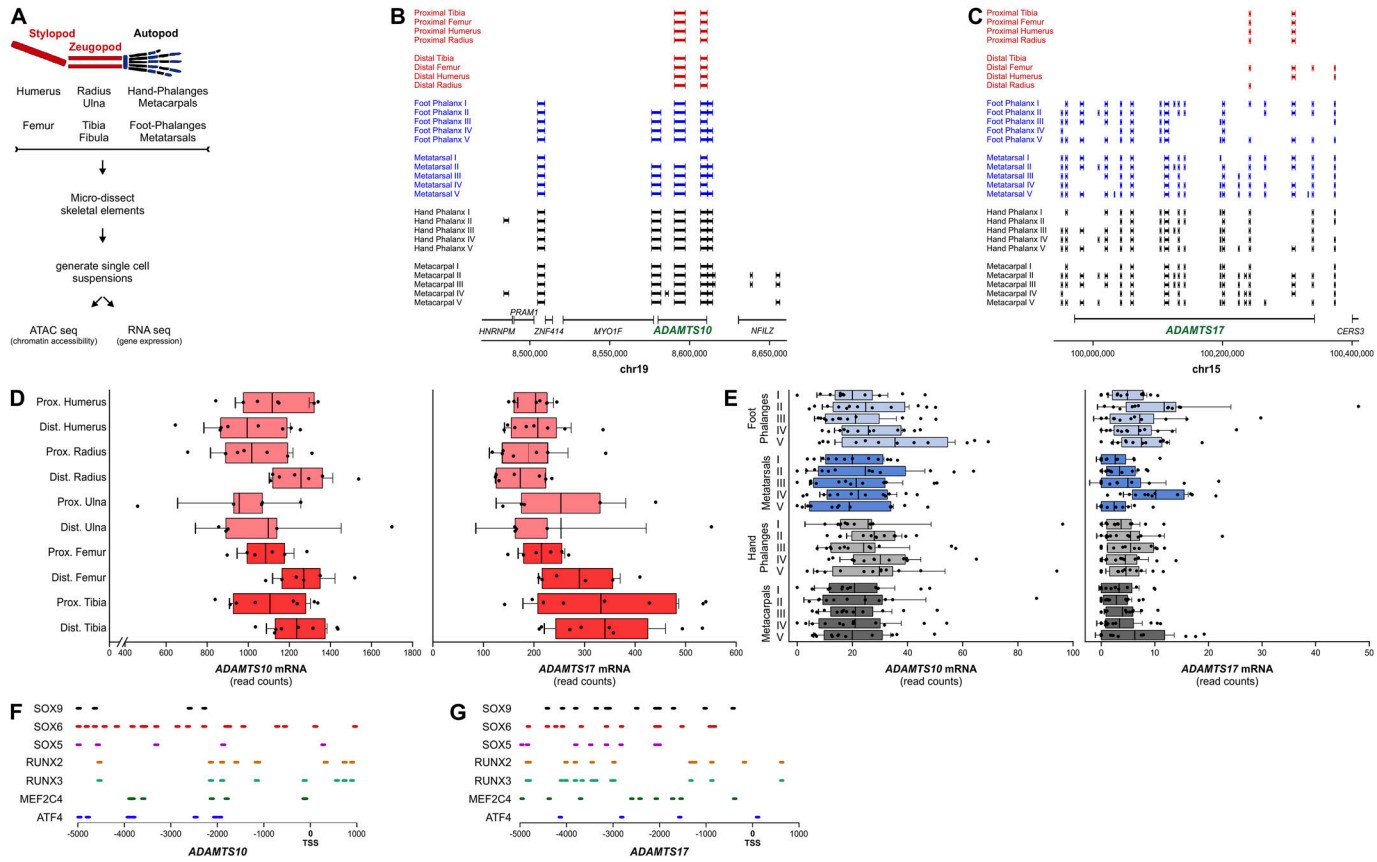

**Figure 4. Chromatin accessibility and putative regulation of *ADAMTS10* and *ADAMTS17* in human cartilage and bone development.**
**(A)** Limb skeletal elements and experimental design to generate ATAC-seq and transcriptomics data from skeletal elements of human products from conception (E54&E67). **(B, C)** Mapping of open chromatin regions identified by ATAC-seq 100 kb up- or downstream of *ADAMTS10* (B) and *ADAMTS17* (C). The positions of *ADAMTS10* on human chromosome 19 and *ADAMTS17* on human chromosome 15 are indicated. **(D, E)** Normalized read counts indicating *ADAMTS10* and *ADAMTS17* mRNA abundance in skeletal elements from stylopods and zeugopods (D) and autopods (E). **(F, G)** Location of transcription factor binding sites 5 kb upstream of the *ADAMTS10* (F) or *ADAMTS17* (G) transcriptional start site (TSS).

dermal papilla cells and *Adamts17* was strongly expressed in the bulge stem cell population (Fig 5H). Expression in other cell types, including fibroblasts, was much lower and differential expression of *Adamts10* and *Adamts17* was less apparent. We next validated temporal *Adamts17* expression dynamics in the skin by RNA in situ hybridization and immunostaining of embryonic and postnatal mouse skin. At E13.5, ADAMTS17 mRNA was localized predominantly in the developing epidermis (Fig 5I). As previously described, at E16.5, high ADAMTS17 mRNA levels were observed in the placode and the hair peg of the developing hair follicles, whereas lower *Adamts17* expression was observed in the epidermis and cells of the dermal layer (Fig 5J) (Hubmacher et al, 2017). At P0, ADAMTS17 mRNA was concentrated in cells surrounding the base of the hair follicle and in the outer root sheath (Fig 5K). ADAMTS17 immunostaining in postnatal skin confirmed ADAMTS17 protein localization in or around hair follicles and the hair shaft (Fig 5L).

Finally, we investigated fibrillin-1 and fibronectin ECM deposition in primary skin fibroblasts isolated from WT, *Adamts10* KO, *Adamts17* KO, or *Adamts10;Adamts17* DKO mice. Skin fibroblasts from DKOs did not appear to form a fibronectin network and

showed abnormal intracellular accumulation of fibrillin-1 immunoreactivity (Fig 5M and N). Fibroblasts isolated from individual *Adamts10* KO or *Adamts17* KO mice showed intermediate phenotypes with a reduction of fibronectin in both KOs and a reduction of fibrillin-1 in the *Adamts17* KO. Whereas fibronectin in *Adamts17* KO fibroblasts was present in globular structures or short fibers on the cell surface or in the vicinity of fibroblasts, fibronectin in *Adamts10* KO fibroblasts was largely organized in an extracellular fibrillar network. Notably, in areas of *Adamts10* KO fibroblasts, ECM with sparse to no fibronectin network, intracellular fibrillin-1 accumulation was more prevalent. Because DKO fibroblasts were present at reduced cell numbers, the lack of fibronectin and subsequently fibrillin-1 ECM deposition could be due to reduced cell densities, because it is known that fibrillin-1 deposition is dependent on cell density.

## Identification of fibronectin and COL6 as ADAMTS17 binding partners

It was previously reported that ADAMTS10 constitutively had poor protease activity because of a degenerated furin cleavage site

**Table 1.  Accessible genomic regions in *ADAMTS10* in cartilage as determined by ATAC-seq.**

| Gene | Start | End | Skeletal elements |
|---|---|---|---|
| *ADAMTS10* (chr19) | 8483730 | 8486836 | Hand Phalanx II; Metacarpal IV |
| | 8504872 | 8509348 | Foot Phalanges I–V; Metatarsals II–V; Hand Phalanges I–V; Metacarpals I–V |
| | 8576228 | 8582004 | Foot Phalanges II–V; Metatarsals II–V; Hand Phalanges I–V; Metacarpals I–V |
| | 8584785 | 8586747 | Metacarpal IV |
| | 8590420 | 8597323 | Distal and Proximal Femur; Distal and Proximal Humerus; Distal and Proximal Radius; Distal and Proximal Tibia; Foot Phalanges I–V; Metatarsals II–V; Hand Phalanges I–V; Metacarpals I–V |
| | 8606826 | 8611019 | Distal and Proximal Femur; Distal and Proximal Humerus; Distal and Proximal Radius; Distal and Proximal Tibia; Foot Phalanges I–V; Metatarsals I–V; Hand Phalanges I–V; Metacarpals I–V |
| | 8611370 | 8614416 | Foot Phalanges I, II, IV, V; Metatarsals II, III, V; Hand Phalanges I–V; Metacarpals I–V |
| | 8614566 | 8615966 | Metacarpal II, III |
| | 8637955 | 8639246 | Metacarpal II, III |
| | 8654156 | 8656442 | Metatarsal III; Hand Phalanx V; Metacarpals II, III, V |

The start and end sites of the open chromatin regions were mapped to chromosome 19 (*ADAMTS10*) based on human genome assembly GRCh38 (hg38).

(GLKR instead of a canonical RX[K/R]R↓ site), which restricted furin-mediated activation (Fig 6A) (Kutz et al, 2011; Wang et al, 2019). In contrast, ADAMTS17 is an active protease based on extensive auto-proteolysis at the cell surface (Hubmacher et al, 2017). To identify additional ADAMTS17 substrates, we used an unbiased mass spectrometry (MS)-based N-terminomics strategy, terminal amine isotopic labeling of substrates (TAILS) (Kleifeld et al, 2011). We complemented this approach by yeast-2-hybrid protein–protein interaction screening. For TAILS, we cocultured human dermal fibroblasts with HEK293 cells stably expressing ADAMTS17 or its active site mutant ADAMTS17-EA (Glu-390 to Ala), which abolishes its autocatalytic activity (Fig 6B, left) (Hubmacher et al, 2017). After isobaric tag labeling of the samples, we identified differentially abundant N-terminally labeled peptides uniquely present or elevated in ADAMTS17 conditioned medium and prioritized the peptides with neo-N-termini from secreted and/or ECM proteins as candidate substrates (Fig 6C). Among potential ADAMTS17 substrates, we identified peptides from COL1A1, COL6A2, COL6A3, and fibronectin (FN1). Multiple ADAMTS17 peptides were identified in the WT ADAMTS17 samples because of its auto-catalytic activity and served as positive controls. In parallel, we used the ADAMTS17 ancillary domain (17-AD) as the bait in a yeast-2-hybrid screen to identify binding partners. This approach identified the ECM proteins thrombospondin-1 (THSB1) and fibulin-3 (FBLN3), the secreted proteases ADAM12 and PAPPA, and the extracellular domain of the catalytically inactive receptor tyrosine protein kinase ERBB3 (Fig 6D). Most notably, we also identified fibronectin and COL6A2 as potential ADAMTS17 binding partners, which overlapped with the results from the MS screen. Therefore, we selected fibronectin and COL6 for further investigation and validation as potential ADAMTS17 substrates.

Based on the yeast-2-hybrid data, the ADAMTS17 ancillary domain interacted with the C-terminal region of fibronectin comprising FNIII domain #15 and FNI domains #10–12 (amino acid residues 2,130–2,416, NP_997647) (Fig 6E and F). The N-terminally labeled fi-bronectin peptide identified in the MS approach localized to the same region (amino acid residues 2,282–2,317) and covered the linker region between FNIII #15 and FNI #10 and the N-terminal part of FNI #10. For biochemical validation of fibronectin as an ADAMTS17 substrate, we first used Western blot of conditioned medium and cell lysates collected from human dermal fibroblasts cocultured with ADAMTS17- or ADAMTS17-EA–expressing HEK293 cells equivalent to the MS approach (Fig 6B, left). Using five different antibodies against fibronectin, we detected distinct fibronectin fragmentation patterns in conditioned medium in the presence of ADAMTS17-, but not ADAMTS17-EA–expressing HEK293s (Fig 6G). All antibodies were raised against full length plasma fibronectin, except 15613-1-AP (ProteinTech), which was raised against a region comprising FNIII domain #15 and FNI domains #10–12. Fibronectin fragmentation in the cell lysate, which included the ECM fraction, was not observed. Together, this suggested that ADAMTS17 protease activity correlated with fibronectin fragmentation. To directly test if ADAMTS17 can cleave fibronectin, we co-transfected HEK293 cells with ADAMTS17- or ADAMTS17-EA–encoding plasmids and a plasmid encoding V5-tagged fibronectin (rFN) (Fig 6B, right). When we analyzed conditioned medium and cell lysates harvested after co-expression of these plasmids, we did not detect fibronectin fragmentation in the medium or cell lysate with a polyclonal antibody against fibronectin or an antibody against the V5 tag of rFN (Fig 6H). This suggested that fibronectin may not be a direct ADAMTS17 substrate or that ADAMTS17 selectively cleaved fibronectin fibrils, which are assembled by dermal fibroblasts.

For COL6A2, the yeast-2-hybrid data suggested an interaction region for 17-AD comprising the C-terminal portion of the central triple helical domain and an N-terminal portion of the von

**Table 2. Accessible genomic regions in *ADAMTS17* in cartilage as determined by ATAC-seq.**

| Gene | Start | End | Skeletal elements |
|---|---|---|---|
| *ADAMTS17* (chr15) | 99950945 | 99953010 | Foot Phalanges II–V; Metatarsals I–V; Hand Phalanges III–V; Metacarpals II–V |
| | 99957622 | 99960241 | Foot Phalanges I–III, V; Metatarsals I–V; Hand Phalanges I, III, IV, V; Metacarpals I–III, V |
| | 99980194 | 99985410 | Foot Phalanges I–III, V; Metatarsals I, II, V; Hand Phalanges III, V; Metacarpals II, III, V |
| | 100007670 | 100009415 | Foot Phalanx II; Metatarsal II; Hand Phalanx IV; Metacarpals II, V; |
| | 100018494 | 100022360 | Foot Phalanges I–III, V; Metatarsals I–III, V; Hand Phalanges I, III, IV, V; Metacarpals II, III |
| | 100032576 | 100033757 | Metatarsal V |
| | 100042238 | 100044322 | Foot Phalanges I–V; Metatarsals I–V; Hand Phalanges I–V; Metacarpals I–V |
| | 100058266 | 100062127 | Foot Phalanges I–V; Metatarsals I–V; Hand Phalanges I–V; Metacarpals I–V |
| | 100103369 | 100105117 | Foot Phalanges I–V; Hand Phalanges I–V |
| | 100110533 | 100116846 | Foot Phalanges I–V; Metatarsals I–V; Hand Phalanges I–V; Metacarpals I–V |
| | 100124861 | 100126574 | Foot Phalanx I; Metatarsal II; Hand Phalanx III; Metacarpals II, III |
| | 100131195 | 100133136 | Foot Phalanges I, II; Metatarsals I–V; Hand Phalanges I–V; Metacarpals I–V |
| | 100140773 | 100142360 | Foot Phalanges I, II; Metatarsals I, II, IV, V; Hand Phalanges I, II, V; Metacarpals I-III, V |
| | 100196165 | 100197462 | Foot Phalanges III, V; Metatarsals I, IV, V; Hand Phalanges I, III, IV, V; Metacarpals I-III, V |
| | 100199818 | 100203168 | Foot Phalanges I–V; Metatarsals II–V; Hand Phalanges I–V; Metacarpals I–V |
| | 100224224 | 100226042 | Metatarsals III–V; Hand Phalanx V; Metacarpals II-V |
| | 100234080 | 100236829 | Metacarpal II, IV, V |
| | 100241067 | 100243238 | Distal and Proximal Femur; Proximal Humerus; Distal and Proximal Radius; Proximal Tibia; Foot Phalanges I, V; Metatarsals I–V; Hand Phalanx I–V; Metacarpals I–IV |
| | 100264745 | 100266902 | Foot Phalanges I, II; Metatarsals I, II, IV, V; Metacarpal V |
| | 100307139 | 100312190 | Distal and Proximal Femur; Distal and Proximal Humerus; Proximal Radius; Proximal Tibia; Foot Phalanges I, II; Metatarsals I, II, IV, V; Hand Phalanx V; Metacarpals II–IV |
| | 100331806 | 100332722 | Metatarsal V |
| | 100338262 | 100340520 | Distal Femur; Foot Phalanges I, II, V; Metatarsals I–V; Hand Phalanges I–V; Metacarpals II, III, V |
| | 100372584 | 100374300 | Distal Femur; Distal Humerus; Foot Phalanges I–III, V; Metatarsals I–V; Hand Phalanges I, II, IV, V; Metacarpals I–V |

The start and end positions of the open chromatin regions were mapped chromosome 15 (*ADAMTS17*) based on human genome assembly GRCh38 (hg38).

Willebrand factor A (VWA) domain #C1 (amino acid residues 480–652, NP_001840.3) (Fig 7A). TAILS identified an N-terminally labeled peptide originating from COL6A3 (amino acid residues 1,348–1,367, NP_004360.2), which was localized in the C-terminal portion of VWA domain #N4 (Fig 7B and C). For validation of COL6 as ADAMTS17 substrate, we used the cell culture setups shown in Fig 6B. In the coculture system of ADAMTS17-expressing HEKs with HDFs, we did not observe a different pattern of bands originating from endogenous COL6 in conditioned medium in the presence of active ADAMTS17 as detected with a COL6A1 antibody (Fig 7D). However, we noticed the disappearance of a ~125 kD COL6A1-reactive band in the cell lysate/ECM fraction when proteolytically active ADAMTS17 was present. When visualizing COL6 ECM deposition in this coculture system by immunostaining, we observed reduced COL6A1 staining in the presence of ADAMTS17 compared with ADAMTS17-EA (Fig 7E and F). We observed a similar difference when

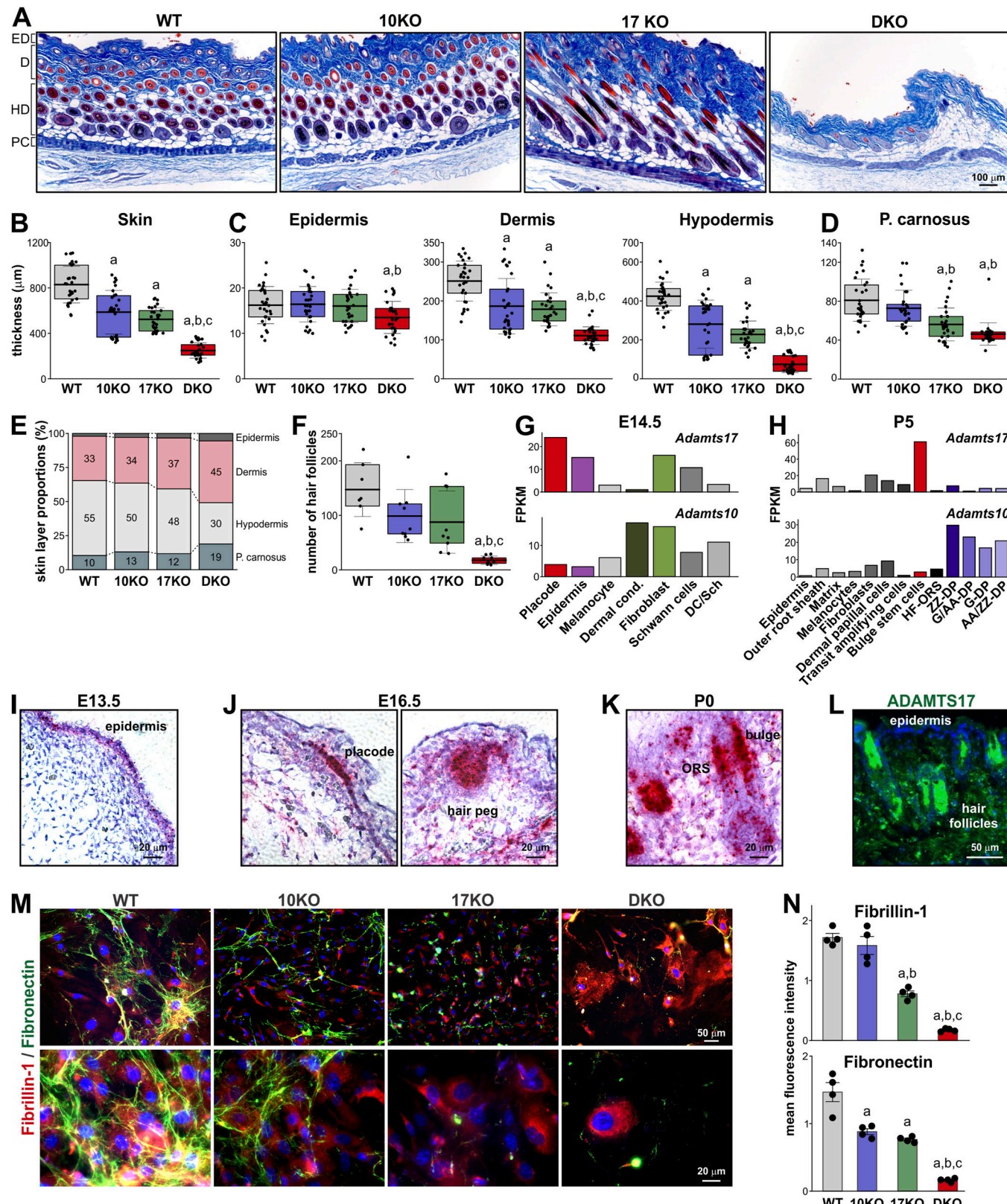

**Figure 5. Adamts10;Adamts17 DKO is associated with skin alterations and aberrant ECM deposition by dermal fibroblasts.**
**(A)** Micrographs of Masson's trichrome–stained cross sections through dorsal skin from 4-wk-old WT, *Adamts10* KO (10KO), *Adamts17* KO (17KO), and DKO mice. ED, epidermis; D, dermis; HD, hypodermis; PC, panniculus carnosus. **(B, C, D)** Bar graphs showing quantification of overall skin thickness (B) and the thicknesses of the epidermis, dermis, hypodermis (C), and panniculus carnosus (p. carnosus, (D)). Individual data points represent multiple measurements along the different skin layers

fibroblasts were cultured in the presence of cell-free conditioned medium collected from ADAMTS17- or ADAMTS17-EA–expressing HEK293 cells. In the presence of ADAMTS17, COL6A1 in the ECM was lower than the amount of COL6A1 deposited in the presence of ADAMTS17-EA (Fig 7G and H). These observations would be consistent with proteolytically active ADAMTS17 limiting the amount of COL6 deposited in the ECM, potentially via proteolysis, but could also be explained by inactive ADAMTS17-EA promoting COL6 deposition into the ECM. To determine, if COL6 is a direct ADAMTS17 substrate, we co-expressed FLAG-tagged recombinant (r)COL6A2 with ADAMTS17 or ADAMTS17-EA in HEK293 cells. Using Western blot detection of the FLAG-tag, we did not observe rCOL6A2 fragmentation in the medium or cell lysate and only detected full length COL6A2 (Fig 7I). However, we noticed an increase in the band intensity for COL6A2 in the ADAMTS17-EA lysate, which could be the result of decreased proteolysis, increased cellular retention, or cell surface/ECM association. To determine, if ADAMTS17 co-localized with COL6, we cultured fibroblasts producing endogenous COL6, in the presence of the previously described purified recombinant catalytic ADAMTS17 domains (17-PCD) or its ancillary domains (17-AD) and co-immunostained for endogenous COL6A1 and recombinant ADAMTS17 (α-Myc) (Hubmacher et al, 2017). Endogenous COL6 in the ECM of fibroblasts costained with both ADAMTS17 protein constructs, suggesting the possibility of an interaction in the ECM (Fig 7J). Finally, we analyzed endogenous COL6 distribution and ECM deposition in dermal fibroblasts isolated from a patient with WMS due to a ADAMTS17 Thr343Ala mutation, where we previously showed intracellular retention and reduced ECM deposition of fibronectin, fibrillin-1, and COL1 (Karoulias et al, 2020a). Compared with control adult human dermal fibroblasts, we observed a strong reduction of COL6 ECM deposition in WMS dermal fibroblasts and concurrent intracellular COL6 accumulation (Fig 7K and L). This observation was confirmed by Western blot analysis, where COL6A1 was decreased in the medium from WMS patient–derived dermal fibroblasts and increased in the cell lysate compared with adult human dermal fibroblasts (Fig 7M and N). Collectively, our data suggested that fibronectin and COL6 are potential ADAMTS17 binding partners and/or substrates, which require further validation.

## Discussion

The fact that mutations in *ADAMTS10* and *ADAMTS17* can both cause short stature in WMS implicates both genes in the regulation of bone growth likely via an impact on growth plate function. Because

it is unclear whether both genes interact in this process, we analyzed the phenotypes of *Adamts10;Adamts17* DKO mice. We showed that combined inactivation of *Adamts10* and *Adamts17* exacerbated bone shortening when compared with individual KOs and compromised overall postnatal survival. We speculate that postnatal lethality in DKO mice is due to compromised heart valve or lung function. Pulmonary, aortic, and mitral valve stenosis, as well as mitral valve regurgitation have all been reported in patients with WMS, which could be exacerbated when ADAMTS10 and ADAMTS17 are both absent (Marzin et al, 2023). Even though lung function appears to be normal in WMS patients, ADAMTS10 and ADAMTS17 are expressed in murine lung tissue and possibly could compensate for each other in individual KOs and WMS, but not in the DKO (Hubmacher et al, 2017; Wang et al, 2019). In addition, we provide evidence that ADAMTS10 and ADAMTS17 are required for skin development, which may relate to skin alterations described in WMS patients (Marzin et al, 1993, 2023). Finally, we identified fibronectin and COL6 as potential ADAMTS17 substrates, which could point towards functions for ADAMTS17 as a regulator of ECM formation and homeostasis.

Bone growth is largely driven by growth plate activity due to proliferation and hypertrophic expansion of chondrocytes. The disruption of either process can lead to bone shortening and short stature (Breur et al, 1991; Wilsman et al, 1996a, 1996b; Cooper et al, 2013). The reduction in the width of the hypertrophic zone observed in the DKO growth plate could thus be a result of decreased formation and/or accelerated turnover of hypertrophic chondrocytes, which depend on chondrocyte proliferation and matrix metalloprotease (MMP)–mediated ECM remodeling at the ossification front, respectively (Vu et al, 1998; Inada et al, 2004; Kennedy et al, 2005). Key enzymes that regulate these processes include MMP13 and MMP9, which can cleave COL2 and COL10 and promote the vascularization of the growth plate (Vu et al, 1998; Inada et al, 2004). When MMP13 or MMP9 were inactivated, the length of the hypertrophic zone in developing bones was increased by ~70% and primary ossification was delayed, indicating a delay in chondrocyte turnover, in particular chondrocyte apoptosis (Vu et al, 1998; Inada et al, 2004). A shorter hypertrophic zone as observed in DKO growth plates could indicate that ADAMTS10 and ADAMTS17 may attenuate chondrocyte turnover potentially through regulating the activity of MMP13 or MMP9 (Knauper et al, 2002). It is interesting to note that ADAMTS10 WMS knock-in and ADAMTS17 KO mice show opposite growth plate phenotypes, each resulting in reduced bone length. The knock-in of the ADAMTS10 Ser236X mutation (Arg237X in humans) resulted in an expansion of the hypertrophic zone, which correlated with a 6–10% reduction in long bone length (Mularczyk et al, 2018). In contrast, inactivation of *Adamts17* resulted in a

from n = 3 mice/genotype. **(E)** Stacked bar graphs showing the relative proportions of individual skin layers. The percentage values are indicated. **(F)** Bar graphs show the quantification of hair follicle numbers in the skin for each genotype. **(G, H)** Bar graphs showing normalized gene expression in fragments per kilobase of transcript per million mapped reads (FPKM) for *Adamts10* and *Adamts17* in individual skin cell types at E14.5 (G) and P5 (H). Data were extracted from the Hair-GEL database (Sennett et al, 2015; Rezza et al, 2016). **(I, J, K)** Micrographs showing the localization of ADAMTS17 mRNA (red/dark purple) in WT skin cross sections at E13.5 (I), E16.5 (J), and P0 (K) detected by RNAscope in situ hybridization with a probe specific for ADAMTS17 mRNA. Sections were counterstained with hematoxylin. **(L)** Micrograph of ADAMTS17 immunostaining (green) of cross sections through WT skin. Nuclei were stained with DAPI (blue). **(M)** Micrographs of primary mouse skin fibroblasts after immunostaining for fibrillin-1 (red) and fibronectin (green). Nuclei were counterstained with DAPI (blue). **(M, N)** Quantification of mean fluorescence intensity from (M) (n = 4 biological replicates). In (B, C, D, F), floating bars indicate 25th–75th percentile range, lines the mean value and whiskers the SD. In (N), the bars represent the mean value and the whiskers the SD. Statistical differences in (B, C, D, F, N) were determined using a one-way ANOVA with post hoc Tukey test. a, *P* < 0.05 compared with WT; b, *P* < 0.05 compared with *Adamts10* KO; *P* < 0.05 compared with *Adamts17* KO.

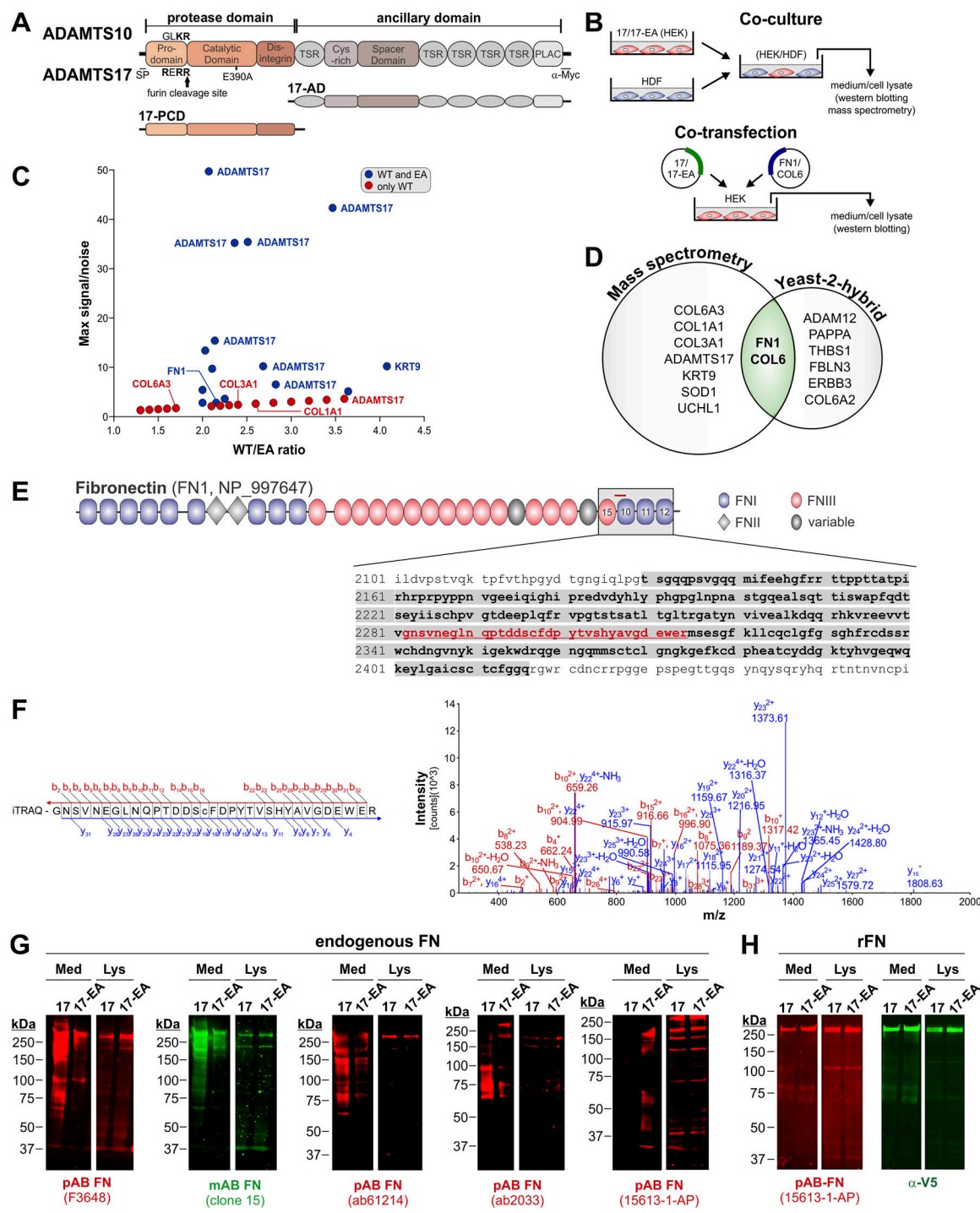

**Figure 6. Identification of fibronectin as a potential ADAMTS17 substrate.**
**(A)** Domain organization of ADAMTS10 and ADAMTS17, which is identical. The degenerate (ADAMTS10) and canonical (ADAMTS17) furin-processing sites and the localization of the catalytic residue Glu-390 in ADAMTS17 (17) that was mutated into Ala to generate proteolytically inactive ADAMTS17-EA (17-EA) are indicated. The domain organization of the catalytic (17-PCD) and ancillary (17-AD) domain constructs is indicated. **(B)** Schematic representation of experimental design for coculture of human dermal fibroblasts (HDF) with HEK293 cells stably expressing 17- or 17-EA (left) or co-transfection of 17- or 17-EA–encoding plasmids with FN1 or COL6A2-encoding plasmids in HEK293 cells (right). **(C)** Volcano plot showing N-terminally labeled peptides identified by N-terminomics method TAILS in conditioned medium from ADAMTS17-expressing HEK293 cells cocultured with HDFs. Peptides present only in samples from WT ADAMTS17 (red) or enriched in conditioned medium from WT ADAMTS17 cocultures compared with the cocultures with proteolytically inactive ADAMTS17-EA suggest ADAMTS17 substrates. **(D)** Venn diagram showing overlap of ADAMTS17-cleaved proteins (TAILS) from coculture systems (left) and binding partners for the ADAMTS17 ancillary domain (17-AD) identified by yeast-2-hybrid screening with a human placenta–derived cDNA library (right). Note that fibronectin (FN1) and COL6 were independently identified in both screens. **(E)** Domain organization of fibronectin (FN1, NP_997647) showing the localization of the domains that interacted with 17-AD (grey box, bolded amino acid sequence) and the localization of the peptide identified by TAILS (red bar, red amino acid sequence). **(F)** MS2 spectrum of the N-terminally labeled FN1 peptide (GNSVNEGLNQPTDDSCFDPYTVSHYAVGDEWER) showing b- and y ions. **(G)** Western blot detection of endogenous fibronectin in conditioned medium (Med) and cell lysates (Lys) collected after coculture of 17- or 17-

shorter hypertrophic zone, also correlating with a 6–10% reduction in long bone length (Oichi et al, 2019). These reports would suggest that ADAMTS10 and ADAMTS17 have distinct roles in regulating growth plate activity, both resulting in shorter bones. This interpretation is to some extent supported by our findings, where mineral deposition was reduced in ADAMTS17-deficient primary rib chondrocytes but was unchanged in ADAMTS10-deficient chondrocytes. The latter findings could be explained by differential compensation, where ADAMTS17 could potentially compensate for the lack of ADAMTS10 in primary chondrocytes, but not vice versa. Alternatively, ADAMTS10 and ADAMTS17 could differentially regulate signaling pathways that regulate growth plate activity and bone growth. In this context, it was shown that reduced BMP signaling in *Adamts17* KO mice delayed terminal differentiation of chondrocytes (Oichi et al, 2019). In the skin of *Adamts10* WMS knock-in mice, BMP signaling, but not TGFβ signaling, was also reduced (Mularczyk et al, 2018). How ADAMTS10- or ADAMTS17 deficiency translates into reduced BMP signaling is not clear but could be secondary to changes in fibronectin and fibrillin-1 deposition, which can regulate extracellular BMP activity (Sengle et al, 2008; Wohl et al, 2016).

Skin thickening is a feature of WMS syndrome, and decreased skin elasticity in *Adamts10* KO and *Adamts17* KO mice as well as increased skin thickness in *Adamts10* WMS knock-in mice were reported previously (Mularczyk et al, 2018; Oichi et al, 2019; Wang et al, 2019). Surprisingly, we noticed skin fragility during routine handling of *Adamts10;Adamts17* DKO mice and, accordingly, observed reduced skin thickness in these mice. One reason for these contrasting observations could be that skin thickening was previously observed in older mice, i.e., at 3 and 8 mo of age in the *Adamts17* KO and at 3 mo of age in the *Adamts10* WMS knock-in (Mularczyk et al, 2018; Oichi et al, 2019). Therefore, it is possible that the skin thickness might still increase in our individual KOs with age, which could be addressed using conditional *Adamts10* and *Adamts17* alleles to overcome lethality. However, thick skin was described in young (21–28 yr) and old (86 yr) WMS patients, suggesting no age-related changes in the WMS skin phenotype (Faivre et al, 2003a; Kutz et al, 2008). Therefore, the skin phenotypes that we have observed remain unexplained.

Because ADAMTS10 and ADAMTS17 are members of the ADAMTS protease family, they are presumed to fulfil their biological function through their respective protease activities and in extension the consequences of substrate cleavage. However, ADAMTS10 is the only ADAMTS protease with a degenerated furin cleavage site (GLKR↓, lacking a canonical Arg residue at the P4 position) and is incompletely activated by furin-mediated removal of the inhibitory prodomain (Kutz et al, 2011). It was shown that ADAMTS10 could cleave fibrillin-1 and fibrillin-2 efficiently after restoring a canonical furin recognition site (RRKR↓) by mutagenesis, suggesting that ADAMTS10 has intrinsic protease activity when activated (Kutz et al, 2011; Wang et al, 2019). In *Adamts10* KO and *Adamts10* WMS knock-in mice, fibrillin-2

accumulation in the ciliary zonule of the eye was observed and could be explained by the absence of such a "fibrillin-ase" activity in ADAMTS10-deficient tissue. Indeed, furin activated ADAMTS10 cleaves fibrillin-2 (Wang et al, 2019). However, fibrillin-2 accumulation could also be explained by ADAMTS10 promoting the assembly of fibrillin-1, which would be reduced in the *Adamts10* KO and result in increased fibrillin-2 exposure to antibodies in the fibrillin microfibril bundles of the ciliary zonule (Kutz et al, 2011; Mularczyk et al, 2018; Wang et al, 2019). Alternatively, ADAMTS10 could be involved in promoting the switch from developmental fibrillin-2-rich microfibrils to postnatal fibrillin-1-rich microfibrils, which similarly would be lacking or be reduced in ADAMTS10-deficient tissues. Therefore, it is plausible that ADAMTS10 may primarily regulate ECM formation both via protease-dependent and protease-independent activities on fibrillin-2 and fibrillin-1, thus modulating the formation and isoform composition of fibrillin microfibrils.

For ADAMTS17, we showed that it cleaves itself at multiple sites, including in the active site, and is thus proteolytically active (Hubmacher et al, 2017). To identify potential additional substrates, we used a complementary MS and yeast-2-hybrid approach. In both screens, fibronectin and COL6 were identified as potential substrates and binding partners for ADAMTS17. Since we did not observe direct proteolysis for both putative substrates, we suggest that ADAMTS17 may play a role in regulating fibronectin and COL6 secretion or ECM assembly either directly or indirectly. Consistent with this interpretation, we reported dysregulated secretion and/or ECM assembly of fibronectin, fibrillin-1, and COL1 in WMS patient–derived skin fibroblasts harboring an *ADAMTS17* mutation (Karoulias et al, 2020a). Since fibronectin is an early scaffolding protein for other ECM proteins, including fibrillin-1 and COL1, it could be that the effect on fibrillin-1 ECM deposition that we have observed here and reported in our previous study is a consequence of compromised fibronectin secretion or ECM assembly in the absence of ADAMTS17. The observation that exogenous ADAMTS17, but not its active site mutant form, reduced COL6 ECM deposition in fibroblast is more consistent with an inhibitory role for ADAMTS17 in this context. In humans, mutations in COL6 genes cause muscular dystrophies and myopathies with variable severity and age of onset (Lamande & Bateman, 2018). These patients suffer from the consequences of low muscle mass and progressive loss of muscle function. This contrasts with WMS and other acromelic dysplasias, where a (pseudo)muscular build, i.e., increased muscle mass, is reported in 68–70% of patients, independent of the genetically defined WMS subtype (Faivre et al, 2003a). How these findings can be reconciled is currently unclear. It is worth mentioning that opposite muscle phenotypes have been observed in patients with mutations in the same gene, notably in *FBN1*, where some mutations cause WMS, featuring a (pseudo) muscular built, whereas most *FBN1* mutations cause Marfan syndrome, which is characterized by low muscle mass (Sakai & Keene, 2019). Therefore, it is possible that the phenotypic

EA–expressing HEKs with HDFs. A monoclonal (green) and four different polyclonal (red) anti-fibronectin antibodies were used. **(H)** Western blot detection of recombinant fibronectin (rFN) in conditioned medium (Med) and cell lysate (Lys) collected after co-expression of 17 or 17-EA with rFN in HEK293 cells. A polyclonal anti-fibronectin antibody (red) and a monoclonal anti V5-tag antibody (green) were used to detect rFN.

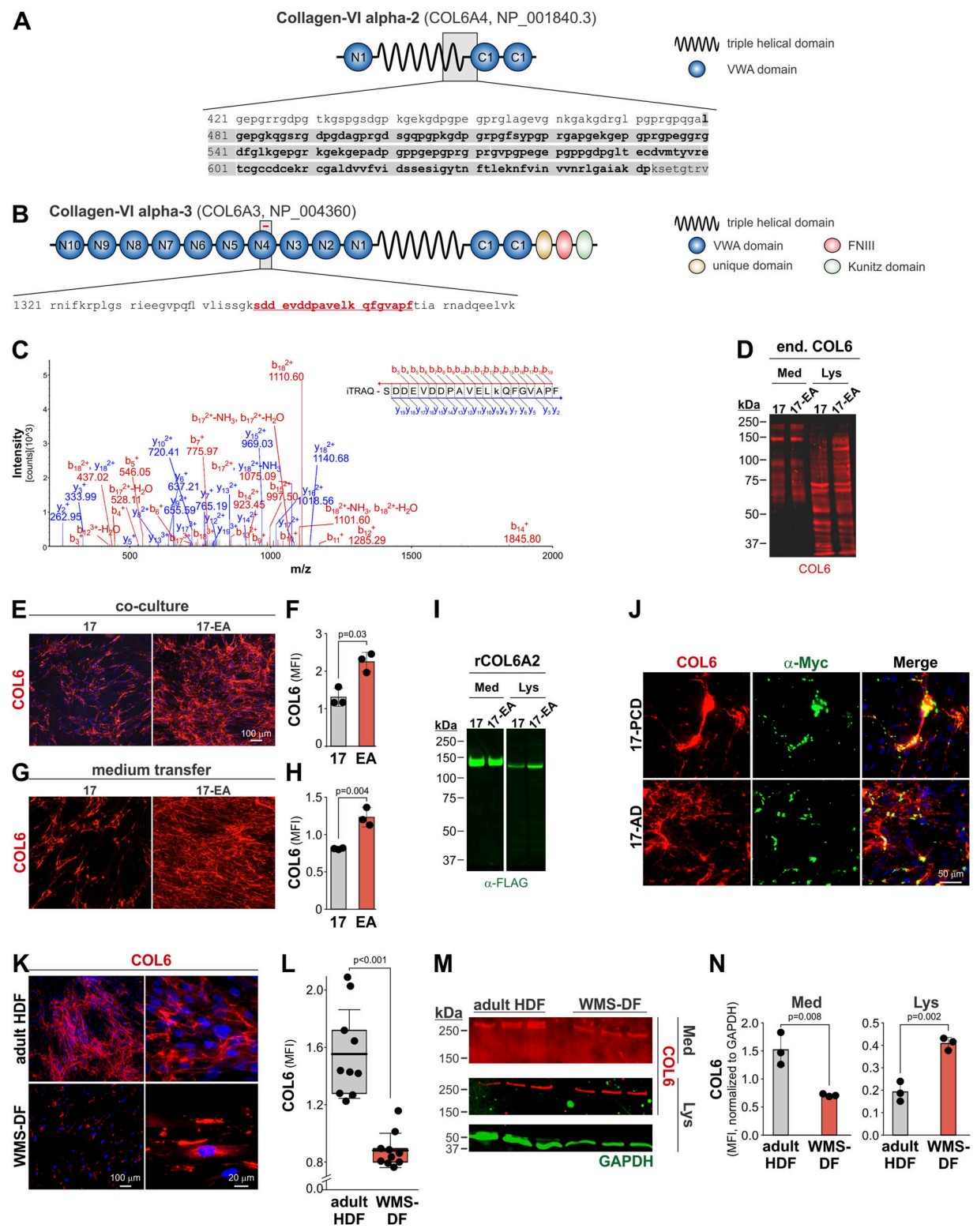

**Figure 7. Identification of collagen VI (COL6) as an ADAMTS17 interacting protein and potential substrate.**
**(A)** Domain organization of COL6A2 (NP_0018403) showing the localization of the domains that interacted with 17-AD (grey box, bolded amino acid sequence).
**(B)** Domain organization of COL6A3 (NP_004360) showing the localization of the peptide identified by MS (red bar, red amino acid sequence). **(C)** MS2 spectrum of the
N-terminally labeled ADAMTS17-digested COL6A3 peptide (SDDEVDDPAVELkQFGVAPF) showing b- and y ions. **(D)** Western blot of endogenous (end.) COL6 (red) in
conditioned medium (Med) and cell lysate (Lys) collected from cocultures of 17- or 17-EA–expressing HEK293 cells with HDFs. **(E)** Micrographs of endogenous
COL6A1 deposition (red) in the ECM of HDFs cocultured with 17- or 17-EA–expressing HEK293 cells. Nuclei were stained with DAPI (blue). **(F)** Quantification of the mean
fluorescence intensity of the COL6A1 signal (n = 3 replicates). **(G)** Micrographs of endogenous COL6A1 deposition (red) in the ECM of HDF after culture in the presence of
conditioned medium from 17- or 17-EA–expressing HEK293 cells. Nuclei were stained with DAPI (blue). **(H)** Quantification of the mean fluorescence intensity of the

consequences of dysregulated COL6 in WMS are different compared with mutations in COL6 genes or that the muscle phenotype in WMS is driven by the dysregulation of ECM proteins other than COL6.

One limitation of our study is the lack of time course experiments because we did not analyze bone or skin development over time. Therefore, we are unable to tell when bone and skin phenotypes arose. The analysis of older mice is challenging because of early postnatal mortality of most DKO mice. Another limitation is the largely descriptive analysis of growth plate and skin phenotypes. We acknowledge that a more mechanistic analysis could have been more informative. In addition, we did not distinguish between changes in fibronectin and fibrillin-1 gene expression or secretion that could account for the observed differences in ECM deposition in primary KO and DKO skin fibroblasts. Finally, the important question of whether ADAMTS10 and ADAMTS17 work as proteases in vivo remains unanswered. This would require the generation of knock-in mice, where ADAMTS10 and ADAMTS17 protease activity is abolished. Such an approach was informative in distinguishing catalytic and non-catalytic functions, for example, for ADAMTS7 (Mizoguchi et al, 2021).

In summary, we provide evidence that ADAMTS10 and ADAMTS17 may (co-) operate in pathways that regulate bone growth and skin development through regulation of ECM deposition and turnover and suggest potential mechanisms on how ADAMTS17 may modulate ECM formation.

# Materials and Methods

### Mouse models

*Adamts10* KO mice generated by Deltagen Inc. and their genotyping were described previously (Wang et al, 2019). In brief, 41 bp of *Adamts10* exon 5 were replaced with an IRES-lacZ-neo cassette resulting in a frameshift and a premature stop codon triggering nonsense-mediated mRNA decay of the *Adamts10* mRNA. The mice were licensed from Deltagen Inc. (agreement #AGR-17486) and maintained in the C57BL/6 strain. *Adamts17* KO mice were generated in C57BL/6 ES cells by CRISPR/Cas9-induced nonhomologous end joining mutagenesis by contract to Applied StemCell, Inc. A guide RNA targeting *Adamts17* exon 3 (5′-GTCCCTCCACCTCCG-TAGCA-3′) was co-injected with Cas9 protein into blastocysts to generate F0 mice, which were screened for frameshift-causing indels. F0 mice harboring an AT insertion in *Adamts17* exon 3 were identified by Sanger sequencing and used to generate F1 mice with the germline-transmitted *Adamts17* AT insertion. F1 mice

harboring the AT insertion were identified by Sanger sequencing of a PCR product generated with primers adjacent to exon 3 (F primer: 5′-GAGAGCATCTGATCAGACGCAAATGG-3′, R primer: 5′-CATGTGACC-CACAGAGTGTCAGC-3′). *Adamts17* KO mice were rederived Center for Comparative Medicine and Surgery at the Icahn School of Medicine. *Adamts17* Het and KO mice were genotyped using genomic DNA extracted from clipped toes as template for a PCR reaction with the following primers: TS17-F 5′-CAGCAGACAGAAGCACAAGCC-3′ and TS17-R 5′-TAGAATCATGGCCCTGACACC-3′. The resulting PCR product was isolated and submitted for Sanger sequencing using the TS17-F primer (Psomagen). The *Adamts17* mutant was maintained in the C57BL/6 strain. Mice heterozygous for the *Adamts10* KO and *Adamts17* KO alleles were crossbred to generate *Adamts10* Het; *Adamts17* Het mice, which were then crossbred to generate *Adamts10;Adamts17* DKO mice. *Adamts10* Het or *Adamts17* Het mice were crossbred to generate age- and sex-matched WT, *Adamts10* KO, or *Adamts17* KO mice. Mice were used between 4–8 wk of age and data from both sexes were combined. All mouse experiments were approved by the Institutional Animal Care and Use Committee (IACUC) of the Icahn School of Medicine at Mount Sinai (protocol numbers IACUC-2018-0009, PROTO202000259, and TR202300000105).

### X-ray imaging

After euthanasia, intact mouse limbs were imaged using a high-resolution radiographic system (UltraFocus digital X-ray cabinet, Faxitron Bioptics). A 10-mm metal rod was used as a scale to enable quantification of bone length.

### Histology

Limbs from 4-wk-old mice were disarticulated and the tibia and femur cut mid-shaft to dissect the knee joint. Muscle and other soft tissue was removed and the knee was fixed in Z-fix (Electron Microscopy Science) for 48 h. After decalcification in 14% EDTA solution, the knee joints were dehydrated, embedded in paraffin, and sectioned through the middle of the knee. Sections were then stained with hematoxylin and eosin (H&E) for histomorphometry. Proximal tibial growth plates were imaged and the dimensions of the growth plate regions measured at multiple points across the growth plate using ImageJ Fiji (NIH) (Schindelin et al, 2012, 2015). Dorsal skin was first shaved to remove hair, and a full thickness rectangular strip was dissected and flattened on filter paper. The filter paper with the skin was then immersed in Z-Fix Zinc Formalin Fixative for 24 h, processed and paraffin embedded. Cross sections were stained with Masson's trichrome stain (HT15; Sigma-Aldrich) according to the manufacturer's protocol and imaged with a Zeiss

COL6A1 signal (n = 3 replicates). **(I)** Western blot of recombinant COL66A2 (rCOL6) in conditioned medium (Med) and cell lysate (Lys) collected after co-expression of 17 or 17-EA and rCOL6A2 in HEK293 cells using a monoclonal anti FLAG-tag antibody (green). **(J)** Micrographs of HDFs cultured in the presence of 50 μg/ml of purified recombinant 17-PCD and 17-AD protein (see Fig 5A for domain organization) costained for endogenous COL6A1 (red) and the Myc-tag of the recombinant ADAMTS17 protein fragments (green). Nuclei were stained with DAPI (blue). **(K)** Micrographs of adult HDFs and Weill–Marchesani syndrome (WMS) patient–derived dermal fibroblasts (WMS-DF) for endogenous COL6A1 (red). Nuclei were counterstained with DAPI (blue). **(L)** Quantification of the mean fluorescence intensity of the COL6A1 signal (n = 3 replicates, 2–3 fields of view). **(M)** Western blot of endogenous COL6A1 (red) and GAPDH (green) in conditioned medium (Med) and cell lysate (Lys) collected from HDF and WMS-DF cultures. **(N)** Quantification of COL6A1 band mean fluorescence intensities normalized to GAPDH. In (E, G, M), bars represent the mean value and whiskers the SD. In (K), the floating bars indicate the 25th–75th percentile range, the lines the mean value, and whiskers the SD. Statistical differences in (E, G, K, M) were determined using a two-sided *t* test.

AxioImager Z1 Upright/Compound Microscope using the following objectives (numerical aperture in brackets): 10x (0.45), 20x (0.8), 40x (0.95), or 63x (0.95). Skin thickness and the thickness of individual skin layers were measured from micrographs using ImageJ Fiji.

## Cell culture assays

Human embryonic kidney (HEK) 293 cells (CRL-1573) and adult human dermal fibroblasts (HDF, PCS-201-012) were purchased from ATCC. Isolation and characterization of the WMS patient–derived dermal fibroblasts (WMS-DF) harboring the *ADAMTS17* c.1027A>G (p.Thr343Ala) mutation was described previously (Karoulias et al, 2020a). Cells were cultured in DMEM supplemented with 10% FBS, 1% L-glutamine, 100 units/ml penicillin, and 100 mg/ml streptomycin (PSG) (complete DMEM) in a 5% $CO_2$ atmosphere in a humidified incubator at 37°C. Upon reaching confluence, the cells were split in a 1:10 (HEK293) or 1:3 (fibroblasts) ratio. Primary fibroblasts were used up to passage 5–7. HEK293 cells stably expressing ADAMTS17 or ADAMTS17-EA were described previously and maintained in complete DMEM supplemented with geneticin (G418) (Hubmacher et al, 2017). For coculture assays, $2 \times 10^6$ HDFs and ADAMTS17 or ADAMTS17-EA ($4 \times 10^6$ cells total) were combined and seeded on a 10 cm cell culture dish. After reaching confluency, the cell layer was rinsed with PBS and cultured in serum-free DMEM. After 48 h, the conditioned medium was collected, and proteins were precipitated with 10% trichloroacetic acid. The cell layer was lysed in RIPA buffer (0.1% NP40, 0.05% DOC, and 0.01% SDS in PBS). Equal volumes were mixed with 5x reducing SDS loading buffer, boiled at 100°C and separated via SDS–PAGE for Western blotting. For direct ADAMTS17 proteolysis assays, HEK293 cells were co-transfected with ADAMTS17 or ADAMTS17-EA and plasmids encoding FN1 or COL6A2 using Lipofectamine 3000. The plasmid encoding V5-tagged fibronectin was described recently and kindly provided by Dr. Dieter Reinhardt (McGill University, Montreal, Canada) (Lee et al, 2017). The COL6A2-encoding plasmid was purchased from GenScript (OHu18654D). After 24 h, cell layers were rinsed with PBS and cultured in serum-free DMEM for an additional 48 h. Conditioned medium was collected and cleared from cell debris by centrifugation. The cell layer was lysed in RIPA buffer, transferred into an Eppendorf tube and cleared of debris by centrifugation. Equal volumes were combined with 5× reducing SDS loading buffer, boiled at 100°C and separated via SDS–PAGE for Western blotting.

## Western blotting

The proteins in equal volumes of media or cell lysates were separated on polyacrylamide gels using SDS–PAGE and blotted onto polyvinylidene difluoride (PVDF) membranes (Immobilon-FL, Merck Millipore Ltd.) using the semi-dry Bio-Rad trans-Blot Turbo transfer system for 33 min at 25 V (Bio-Rad) or a wet transfer system for 1.5 h at 70 V at 4°C both operated with 25 mM Tris, 192 mM glycine, and 20% methanol as transfer buffer. Membranes were blocked with 5% (wt/vol) milk in TBS (10 mM Tris–HCl, pH 7.2 and 150 mM NaCl) for 1 h at RT and incubated with the following primary antibodies diluted in 5% (wt/vol)

milk in TBS-T (TBS + 0.1% Tween 20) at 4°C overnight: polyclonal antibodies against fibronectin (F3648, 1:1,000; Sigma-Aldrich; ab2033, 1:200; Millipore; 15613-1-AP, 1:1,000; ProteinTech; ab61214, 1:200; Abcam), monoclonal antibody against fibronectin (cl 15, 1:400; Sigma-Aldrich), polyclonal antibody against COL6 (ab6588, 1:500; Abcam), monoclonal antibodies against the V5-tag (mAB, 1:500; Invitrogen), and the FLAG-Tag (M2, 1:500; Sigma-Aldrich), or a monoclonal antibody against GAPDH (1:1,000; Millipore). After incubation with the primary antibody, the membranes were rinsed with TBS-T 3 × 5 min at RT and incubated with IRDye-goat-anti-mouse or goat-anti-rabbit secondary antibodies (1:10,000 in 5% (wt/vol); Jackson ImmunoResearch Laboratories) in TBS-T for 2 h at RT. Membranes were then rinsed 3 × 5 min with TBS-T, once in TBS and imaged using an Azure c600 Western blot imaging system (Azure Biosystems). Band intensities were quantified using the AzureSpot analysis software and normalized to GAPDH.

## Immunostaining of tissue sections and cell layers

Growth plate and skin sections were de-paraffinized and rehydrated followed by protease-mediated antigen retrieval using HistoZyme (Diagnostic BioSystems). After blocking with 5% BSA in PBS, the sections were incubated with a mouse monoclonal antibody against ADAMTS17 that was raised against human ADAMTS17 (3B7, 1:500; Novus Biologicals) in blocking buffer in a humidified chamber overnight at 4°C. The sections were rinsed in PBS 3 × 10 min at RT and incubated in AlexaFluor 488–labeled secondary goat–anti-mouse antibody in blocking buffer at for 1 h at RT. The slides were cover-slipped with ProLong Antifade Gold mounting medium and imaged using a Zeiss Axio Observer Z1 fluorescence motorized microscope with definite focus.

HEK293 cells and HDFs or cocultures thereof, or primary chondrocytes were seeded in eight-well chamber slides (50,000 cells/well) (CELLTREAT Scientific Products) and cultured in complete DMEM for 3–4 d. For the analysis of COL6, complete DMEM was supplemented with 100 $\mu$M ascorbic acid (Thermo Fisher Scientific). After medium removal, the cell layers were rinsed with PBS and fixed for 5 min with 150 $\mu$l ice-cold 70% methanol/30% acetone (Thermo Fisher Scientific), which permeabilized the cells, or with 4% PFA for 20 min to only stain the cell surface and ECM proteins, followed by permeabilization with 0.1% Triton X-100 before incubation with the secondary antibody. After fixation, the cells were rinsed with PBS and blocked with 10% normal goat serum (Jackson ImmunoResearch Laboratories) in PBS for 1 h at RT. Chondrocytes were treated with 50 nM okadaic acid (#459620; Sigma-Aldrich), and the controls were treated with DMSO for 24 h and allowed to differentiate. For immunostaining, the cells were fixed with 4% PFA without permeabilization. The cells were incubated with primary antibodies against fibrillin-1, fibronectin, COL6, anti-Myc-tag (MYC tag polyclonal antibody, #16286-1-AP; Proteintech), or ADAMTS17 diluted in blocking buffer overnight at 4°C. The cells were rinsed 3 × 5 min with PBS and incubated with AlexaFluor labeled secondary goat–anti-mouse or goat–anti-rabbit antibodies (1:350 in blocking buffer) (Jackson ImmunoResearch Laboratories) for 2 h at RT followed by 3 × 5 min

rinses with PBS and mounting with ProLong Gold Antifade Reagent with DAPI (Thermo Fisher Scientific). The slides were imaged using a Zeiss Axio Observer Z1 fluorescence motorized microscope with definite focus, and Zeiss Zen Microscope Software and ImageJ Fiji were used to quantify fluorescence pixel intensities.

### mRNA isolation and quantitative real-time PCR

Chondrocyte pellets were removed at experimental time points, immersed in TRIzol reagent and RNA was extracted according to the manufacturer's instructions. Pellets were lysed by pipetting the TRIzol up and down several times to ensure complete homogenization. The lysate was collected into sterile tubes and incubated at room temperature for 5 min. After lysis, 0.2 ml chloroform was added per 1 ml of TRIzol and the Eppendorf tubes were inverted several times and incubated at RT for 2–3 min. To separate the organic and inorganic phase, the samples were centrifuged at 12,000$g$ for 15 min at 4°C. The aqueous phase containing RNA was carefully transferred to a new Eppendorf tube and the RNA was precipitated by adding 0.5 ml of isopropanol per 1 ml of TRIzol reagent and samples were incubated for 10 min at RT. The RNA was pelleted by centrifugation at 12,000$g$ for 10 min at 4°C. The supernatant was discarded, and the RNA pellet was washed with 1 ml 75% ethanol per 1 ml of TRIzol, followed by centrifugation at 7,500$g$ for 5 min at 4°C. The RNA pellet was air dried for 5–10 min at RT and dissolved in 30–50 $\mu$l of RNase-free water depending on pellet size. RNA concentration and purity were measured using a NanoDrop OneC spectrophotometer (Thermo Fisher Scientific). RNA preparations were further purified using DNAse I to remove traces of DNA. Reverse transcriptase was used to convert 1 $\mu$g of RNA into cDNA.

Quantitative real-time (qRT)-PCR was performed in triplicates in a 384-well plate format. For each reaction, 2 $\mu$l cDNA, 0.5 $\mu$l of forward and reverse primers (100 $\mu$M stock), and SYBR Green PCR Master Mix (Applied Biosystems) were combined in a total reaction volume of 10 $\mu$l. The amplification and detection were performed on an ABI PRISM 7900HT Sequence Detection System (Applied Biosystems). All reactions were run under standard cycling conditions. The following PCR primer pairs were used: GAPDH (F: 5′-AGGTCGGTGTGAACGGATTTG-3′, R: 5′GGGGTCGTTGATGGCAACA-3′ or F: 5′-CTTTGTCAAGCTCATTTCCTGG-3′, R: 5′-TCTTGCTCAGTGTCCTTGC-3′), ADAMTS10 (F: 5′- CCCGCCTATTCTA-CAAGGTGG-3′, R: 5′- GCCTTCCCGTGTCCAGTATTC-3′ or F: 5′-GTAGTG-GAGTGCCGAAATCAG-3′, R: 5′-CAGCGTGACCAGTTTCCTAC-3′), ADAMTS17 (F: 5′- CTGCTGTATTTGTGACCAGGAC-3′, R: 5′- AGCACACATTTCCTCTTAGCAC-3′ or F: 5′-CCTTTACCATCGCACATGAAC-3′, R: 5′-ATTCCGTCCTTTTACCCACTC-3′), FBN1 (F: 5′- GGACAGCTCAGCGGGATTG-3′, R: 5′- AGGACACATCTCA-CAGGGGT-3′), FN1 (F: 5′- GCTCAGCAAATCGTGCAGC -3′, R: 5′- CTAGG-TAGGTCCGTTCCCACT -3′), COL2A1 (F: 5′-GGGAATGTCCTCTGCGATGAC-3′, R: 5′-GAAGGGGATCTCGGGGGTT-3′), COL10A1 (F: 5′- TTCTGCTGCTAATGTTCTT-GACC-3′, R: 5′- GGGATGAAGTATTGTGTCTTGGG-3′).

### Isolation and differentiation of primary chondrocytes

Primary costal chondrocytes were isolated as described previously (Mirando et al, 2014). The ribcage of P5 mouse pups was dissected, flattened, and the soft tissue was removed. The cartilaginous portions of the ribs were then transferred into PBS containing 10× penicillin and streptomycin (Gibco). The ribs were digested with 15 ml pronase (2 mg/ml; Sigma-Aldrich) in a 50-ml Falcon tube for 1 h at 37°C in a tissue culture incubator under a 5% $CO_2$ atmosphere. The pronase solution was replaced with 15 ml of collagenase D (3 mg/ml; Sigma-Aldrich) followed by incubation for 1 h under the same conditions with gentle agitation every 10 min. After 45 min of incubation, the collagenase D solution was vigorously pipetted up and down over the thoracic cages. The solution was finally transferred into a 50-ml falcon tube. The soft tissue debris was removed as it sediments slower and the process was repeated with sterile PBS buffer. The cleaned cartilage was then digested again with 15 ml collagenase D solution for 4–5 h at 37°C in the cell culture incubator. Finally, primary chondrocytes were released by pipetting the solution containing the cartilage fragments up and down ~10 times. This digest was passed through a 40-$\mu$m cell strainer and pelleted by centrifugation at 300$g$ for 5 min. The chondrocyte pellet was rinsed twice with PBS and chondrocytes were resuspended in 10 ml complete DMEM medium, plated at a density of $10^5$/cm$^2$ on six-well plates, and cultured in complete DMEM. Upon reaching confluency, the medium was replaced with chondrocyte maturation medium (complete DMEM supplemented with 50 $\mu$g/ml ascorbic acid and 10 mM $\beta$-glycerophosphate). The medium was changed every 2 d, and mineral deposition was visualized after 21 d by alizarin red staining. For this, the differentiated and matured chondrocytes were rinsed with PBS and fixed with 10% formaldehyde (MP Biomedicals) for 15 min at RT. The cell layer was rinsed twice with distilled water and incubated in 1 ml of 40 mM alizarin red staining solution (pH 4.1) per well for 20 min at RT under gentle agitation on a shaker. The plates were incubated at room temperature for 20 min with gentle shaking. The alizarin red staining solution was removed, and the wells were rinsed several times with 5 ml of distilled water. Before bright-field imaging, excess water was removed and the cell layer was dried at RT. The cell layers were imaged, and the stained mineral deposition was quantified using the ImageJ Fiji.

### ATAC-seq and transcriptomics analyses of ADAMTS10 and ADAMTS17 expression

Sample generation, ATAC-seq, RNA sequencing, and data analyses for the data set have been described recently (Richard et al, 2024; Okamoto et al, 2025 *Preprint*). No additional human products of conceptions that were not previously described were used for this study. In brief, epiphyses (long bones) or whole elements (pooled phalanges or metapodials) were microdissected and RNA was extracted after tissue homogenization followed by RNA sequencing. For ATAC-seq, tissues were digested with collagenase to generate single-cell suspensions, which were then subjected to the ATAC-seq protocol. The data sets for the stylopodial and zeugopodial elements are deposited in the Gene Expression Omnibus repository under the accession numbers GSE252289 (human long bone skeleton ATAC-seq) and GSE252288 (human long bone skeleton RNA-seq). The data sets for the

autopodial elements are available under the accession numbers GSE283854 and GSE286924.

## RNAScope in situ hybridization

WT mouse embryos at E13.5, E16.5, and P0 were fixed in 4% paraformaldehyde in PBS overnight at 4°C, dehydrated, and embedded in paraffin. Fresh 6-$\mu$m sections were used for in situ hybridization using RNAscope (Advanced Cell Diagnostics) following the manufacturer's protocol. Tissue localization of ADAMTS17 mRNA was achieved with a probe recognizing the mRNA from mouse *Adamts17* (#316441) in combination with the RNAscope 2.0 HD detection kit "RED." Tissue sections were counterstained with hematoxylin and cover-slipped with Cytoseal 60 (Electron Microscopy Science). After curation of the mounting medium, the sections were observed using an Olympus BX51 upright microscope equipped with a CCD camera (Leica Microsystems) for imaging.

## Isolation of primary mouse skin fibroblasts

Mice were euthanized using $CO_2$ inhalation followed by cervical dislocation and skin fibroblasts were isolated using enzymatic digestion. After euthanasia, the dorsal skin was shaved and a 1–2-$cm^2$ section was excised using sterile scissors and forceps. The excised skin was washed in phosphate-buffered saline (PBS). The skin was minced into 1–2-$mm^2$ fragments and transferred into a 50-ml conical tube. The tissue was digested using 2 mg/ml collagenase type II (LS004202; Worthington) in DMEM. The tube was incubated at 37°C for 1 h with gentle agitation to release the fibroblasts. After digestion, the cell suspension was triturated with a 10-ml pipette to further dissociate the tissue. The suspension was then filtered through a 70-$\mu$m cell strainer to remove undigested fragments. The filtered cell suspension was centrifuged at 300$g$ for 5 min at room temperature. The resulting cell pellet was resuspended in DMEM. The cell suspension was plated onto sterile 10-cm culture dishes and incubated at 37°C, 5% $CO_2$. 24 h post-plating, the medium was replaced to remove non-adherent cells.

## N-terminomics via TAILS

For iTRAQ labeling, we cocultured HDFs with HEK293 cells expressing ADAMTS17 or ADAMTS17-EA (1 × $10^6$ cells each, two 10-cm dishes). After reaching confluence, the cell layer was rinsed with PBS and phenol-free serum-free DMEM was added. After 2 and 4 d, the medium was collected and EDTA (10 mM final concentration) and one tablet of cOmplete EDTA-free protease inhibitors (Roche), dissolved in 250 $\mu$l water, were added. The medium was cleared of cellular debris by centrifugation at 500$g$ for 5 min, sterile filtrated through a 0.22-$\mu$m filter, and stored at −80°C. Media (~40 ml) were thawed and concentrated to 2.5 ml with centrifugal filter devices (Amicon Ultra-4, 3 kD cut-off; #UFC800324). Proteins were then precipitated by adding 20 ml of ice-cold acetone and 2.5 ml ice-cold methanol followed by vortexing and incubation at −80°C for 3 h. Protein precipitates were collected by centrifugation at 14,200$g$ in a Beckman JS-13.1 outswing rotor at 4°C for 20 min. The supernatant was decanted and the protein pellets were air dried. Air-dried pellets

were dissolved in 360 $\mu$l 8 M guanidine-HCl, 504 $\mu$l double-distilled water, and 288 $\mu$l 1 M Hepes buffer resulting in terminal amine isotopic labeling of substrates (TAILS) buffer (final concentration: 2.5 M GnHCl and 250 mM Hepes, pH 7.8).

300 $\mu$g of protein from the TS17X1-WT and TS17X1-mut samples were prepared for iTRAQ-TAILS as previously described (Martin et al, 2020). In brief, protein was reduced and alkylated with DTT and IAA, respectively. Protein samples were mixed at a 1:1 ratio with iTRAQ labels 113 (TS17X1-WT) or 115 (TS17X1-EA) in DMSO at 37°C overnight and quenched with 1 M Tris pH 8. The samples were then combined before overnight digestion with trypsin. N-terminally labeled peptides were enriched as previously described using the hyper-dendritic polyglycerol aldehyde (Nandadasa et al, 2023). Before MS analysis, the samples were desalted using a C18 Ziptip and reconstituted in 50 $\mu$l 1% acetic acid. Peptides were identified with a Dionex Ultimate 3000 UHPLC interfaced with a Thermo LTQOrbitrap Elite hybrid mass spectrometer system. The HPLC column was a Dionex 15 cm × 75 $\mu$m id Acclaim PepMap C18, 2 $\mu$m, 100 Å reversed-phase capillary chromatography column. 5 $\mu$l volumes of the extract were injected and the peptides eluted from the column by an acetonitrile/0.1% formic acid gradient at a flow rate of 0.3 $\mu$l/min were introduced into the source of the mass spectrometer on-line operated at 1.9 kV. The digest was analyzed using a data-dependent MS method acquiring full scan mass spectra to determine peptide molecular weights and product ion spectra to determine amino acid sequence in successive instrument scans. Both collision-induced dissociation (CID) and higher energy collisional dissociation (HCD) fragmentation methods were performed on the top eight most abundant precursor ions in each scan cycle. HCD fragmentation is required to quantify the reporter ions of the iTRAQ labels on peptides. The samples were analyzed using data-dependent acquisition (DDA) using both HCD and CID fragmentations. The resulting data were searched using SEQUEST program which integrated in Proteomics Discoverer 2.5 software package against UniProt human protein sequence database (March 2024). Trypsin (semi) was set as protease, carbamidomethylation of Cys was set as a static modification and iTRAQ 8-plex of peptide N-terminal, Lys and Tyr, oxidation of Met, and N-term Gln cyclization were set as dynamic modifications. Data are available via ProteomeXchange with identifier PXD068317 (Deutsch et al, 2023).

## Yeast-2-hybrid screening

An ULTImate Y2H SCREEN was performed by Hybrigenics Services using ADAMTS17-AD (aa 546–1,122) as bait together with the Prey Library Human Placenta_RP6. The bait was cloned into the pB27 (N-LexA-bait-C fusion) vector. 60 clones were processed and 170 million interactions analyzed.

## Statistical analysis

Statistical analyses were performed using OriginPro 2018 software. N = 2 samples were compared with a two-sided *t* test and n ≥ 3 samples with a one-sided ANOVA. *P*-values < 0.05 were considered statistically different.

# Data Availability

The data sets for the stylopodial and zeugopodial elements are deposited in the Gene Expression Omnibus repository under the accession numbers GSE252289 (human long bone skeleton ATAC-seq) and GSE252288 (human long bone skeleton RNA-seq). The data sets for the autopodial elements were deposited in the Gene Expression Omnibus repository under the accession numbers GSE283854 and GSE286924. The mass spectrometry proteomics data have been deposited to the ProteomeXchange Consortium via the PRIDE partner repository with the data set identifier PXD068317 (Perez-Riverol et al, 2025).

# Acknowledgements

This research was in part supported by a grant from the National Institutes of Health (R01AR070748 to D Hubmacher). We thank Dr. Dieter Reinhardt (McGill University, Montreal, Canada) for kindly providing the fibrillin-1 antibody and the V5-tagged rFN1 expression plasmid. We thank Damien Laudier (Orthopedic Research Laboratories, Icahn School of Medicine at Mount Sinai, New York, NY) for growth plate and skin histology.

## Author Contributions

N Taye: data curation, formal analysis, validation, investigation, visualization, and methodology.
SZ Karoulias: data curation, formal analysis, investigation, visualization, and methodology.
Z Balic: data curation, formal analysis, and investigation.
LW Wang: data curation and investigation.
BB Willard: data curation and formal analysis.
D Martin: data curation, formal analysis, and visualization.
D Richard: data curation, formal analysis, and visualization.
AS Okamoto: data curation, formal analysis, and visualization.
TD Capellini: data curation and formal analysis.
SS Apte: conceptualization, data curation, formal analysis, and investigation.
D Hubmacher: conceptualization, data curation, formal analysis, investigation, visualization, and methodology.

## Conflict of Interest Statement

The authors declare that they have no conflict of interest.

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
