## [Reviewer comments · Life Science Alliance]

Combined ADAMTS10 and ADAMTS17 inactivation exacerbates bone shortening and skin phenotypes

Nandaraj Taye, Stylianos Karoulias, Zerina Balic, Lauren Wang, Belinda Willard, Daniel Martin, Daniel Richard, Alexander Okamoto, Terence Capellini, Suneel Apte, and Dirk Hubmacher

DOI: <https://doi.org/10.26508/lsa.202503232>

Corresponding author(s): Dirk Hubmacher, Icahn School of Medicine at Mount Sinai

Review Timeline:

Submission Date:	2025-01-22
Editorial Decision:	2025-03-06
Revision Received:	2025-08-04
Editorial Decision:	2025-09-11
Revision Received:	2025-09-12
Accepted:	2025-09-17

Scientific Editor: Sarita Hebbar

Transaction Report:

March 5, 2025

Re: Life Science Alliance manuscript #LSA-2025-03232

Dr. Dirk Hubmacher
Icahn School of Medicine at Mount Sinai
Orthopedics
1 Gustave L. Levy Pl
New York 10029

Dear Dr. Hubmacher,

Thank you for submitting your manuscript entitled "Combined ADAMTS10 and ADAMTS17 inactivation exacerbates bone shortening and skin phenotypes" to Life Science Alliance. The manuscript was assessed by expert reviewers, whose comments are appended to this letter. We invite you to submit a revised manuscript addressing the Reviewer comments.

Addressing many of these concerns will require new experimental data to be included and discussed, for example:

- single and double knockout mice comparisons (all three reviewers)
- phenotyping at another age/developmental stage (all three reviewers),
- use of different markers for skin/bone phenotypes (Reviewer 2, comments 1 & 8, Reviewer 3, comment 3)
- corroborating ADAMTS17 immunostaining with additional approaches and appropriate controls (Reviewer 2, comment 3 and Reviewer 3, comment 1).

Thank you for this interesting contribution to Life Science Alliance. We are looking forward to receiving your revised manuscript.

Sincerely,

Sarita Hebbar, PhD

-- By submitting a revision, you attest that you are aware of our payment policies found here: <https://www.life-science->

B. MANUSCRIPT ORGANIZATION AND FORMATTING:

Reviewer #1 (Comments to the Authors (Required)):

Weill-Marchesani is a rare heritable connective tissue disorder, characterized by short stature, brachydactyly, stiff skin and heart and eye anomalies. Autosomal recessive and autosomal dominant forms have been reported. AR can be caused by bi-allelic variants in either ADAMTS10 or ADAMTS17 (among other genes).

In order to gain insights into the potentially overlapping/cooperating pathways into which ADAMTS10 and ADAMTS17 function, the authors generated Adamts10/Adamts17 double knockout (DKO) mice, and studied the growth plate and bone formation as well as ECM organization and skin phenotype of these mice, in comparison with WT, single Adamts10, and single Adamts17 KO mice.

The combined inactivation of Adamts10 and Adamts17 led to significant postnatal lethality compared to single Adamts10 or Adamts17 KO mice. Strikingly Adamts17 gene dosage had a stronger effect and compromised postnatal survival more than Adamts10 gene dosage.

Question: It is not clear however why these mice die. Could the authors speculate on this?

The combined inactivation of Adamts10 and Adamts17 also exacerbated bone shortening when compared to individual KO mice. The severe bone shortening observed in the DKO mice correlated with a narrower hypertrophic zone in the growth plates.

The authors noted fragile skin during routine handling the DKO mice and observed reduced skin thickness in the single and DKO mice (age between 4-8 weeks), with an exacerbated effect in the DKO, which was predominantly driven by a reduction in the hypodermal layer. These findings indeed suggest that ADAMTS10 and ADAMTS17 are required for skin development, which may relate to abnormal skin phenotype described in WMS patients. It is however unclear why the DKO show a fragile, thin skin, which contrasts to the stiff and thickened skin described in WMS patients. Increased skin thickness had already been reported previously in ADAMTS10 and ADAMTS17 single KO, but at later age (3-8 months).

Question: Has an age-related change in skin phenotype (changing from thin, fragile to thick, inelastic skin) been described in human WMS patients?

Comment: I would recommend analyzing the skin in the single and double KO mice at a later time point (e.g. 3 months) to document any changing phenotype.

N-terminomics and yeast two-hybrid screening assays were used to find substrates for ADAMTS17, and the ECM proteins fibronectin and collagen VI were identified as potential substrates, pointing towards a potentially distinct function for ADAMTS17 as a regulator of ECM formation and homeostasis. However, validation experiments did not reveal direct proteolysis of either fibronectin or collagen VI by ADAMTS17. ECM formation was investigated in primary ADAMTS10- and ADAMTS17-deficient skin fibroblasts and this showed abnormal fibronectin deposition, together with aberrant intracellular accumulation of fibrillin-1. These findings indeed support a role for ADAMTS17 in ECM protein secretion and assembly.

Comment: The Identification of fibronectin and Collagen VI as ADAMTS17 binding partners are only briefly discussed in the Discussion Section of the manuscript. This is in imbalance with the other parts. I would recommend expanding this discussion, putting the latter findings (especially about collagen VI) in a broader context.

Question: The finding of a strong reduction of collagen VI ECM deposition in WMS dermal fibroblasts and concurrent intracellular Collagen VI accumulation is very interesting. Is there any evidence that this (or other) WMS patients suffer from a phenotype that overlaps with Ullrich Congenital Muscular Dystrophy, caused by defects in one of the COL6 genes?

This study provides important new insights into the role of ADAMTS10 and ADMATS17 in skin and bone development and

homeostasis. Each of the different sections is well-documented with figures and data that strongly support the conclusions. The manuscript is clearly structured and well-written.

Minor remarks:

- Abstract and introduction:

Conditions (and not variants) are autosomal recessive or autosomal dominant. Variants (or mutations) are bi-allelic or mono-allelic and lead to respectively AR or AD conditions. Please correct this throughout the manuscript

- Page 4 line 66: Homozygosity of a ...variants  variant

- Page 5 line 94: While Adamts10 knockout (KO) mice did not result in short stature  ...did not display short stature

- Page 5, line 104: The findings of our studies, taken together with prior work strongly suggest  The findings of our studies, taken together with prior work, strongly suggest

Reviewer #2 (Comments to the Authors (Required)):

Summary of the Paper

In this study, the authors investigate the role of the secreted ADAMTS proteases, ADAMTS10 and ADAMTS17, using double knockout (dKO) mice. The genetic basis of Weill-Marchesani syndrome (WMS) suggests that altered extracellular proteins contribute to its pathogenesis. Based on this, the authors hypothesize that genetic interactions between ADAMTS10 and ADAMTS17 may be linked to WMS phenotypes, including short stature, short hands, and skin anomalies. As the molecular and cellular mechanisms behind WMS remain unclear, this study provides valuable insights into how these ADAMTS proteases regulate chondrocyte hypertrophy and skin development. Consequently, this work is significant for advancing the field of extracellular matrix biology.

Specific Comments

#1) One weakness of this paper is that the authors only examine bone and skin phenotypes at a single time point (4 weeks of age). Around this age, the survival ratio of ADAMTS10:ADAMTS17 dKO mice is approximately 20% (Fig. 1G). As a result, bone phenotypes (Fig. 2A-N), growth plate analysis (Fig. 3A-C), and skin anomalies (Fig. 5A-H) may be indirectly associated with ADAMTS10 and ADAMTS17 deletion. To clarify this, it would be essential to examine these phenotypes at multiple time points, particularly postnatal day 10, when the survivability of dKO mice is affected. Additionally, the use of established markers (e.g., Col2, ColX immunohistochemistry), cell proliferation assays (e.g., Ki-67 staining, BrdU assay), and cell death assays (e.g., Caspase 3 staining, TUNEL assay) would be valuable to examine the chondrogenesis in ADAMTS10:ADAMTS17 dKO mice.

#2) In Fig. 1C, the authors should use ADAMTS17 KO tissue (rather than dKO) in growth plate and skin fibroblast analysis to examine the reduction of ADAMTS17 expression. The deletion of ADAMTS10 could potentially suppress ADAMTS17 expression indirectly.

#3) In Fig. 3A, presenting low-magnification H&E images would be beneficial. While the authors attribute the bone phenotype to growth plate retardation, ADAMTS17 may also regulate osteogenesis and osteoclastogenesis. To support this, dKO mice show less trabecular bone and/or bone marrow compared with wild-type mice (Fig. 3A, right panel).

#4) To strengthen the authors' interpretation, Safranin O staining should be performed to visualize the morphology of the growth plate.

#5) The ADAMTS17 immunostaining should be improved with appropriate negative controls, as there is a discrepancy between the results shown in Fig. 1C (growth plate in wild-type) and Fig. 3D.

#6) There is a logical disconnect in the use of okadaic acid treatment in primary chondrocytes from ribs to explore the molecular link between ADAMTS10 and ADAMTS17 in chondrocyte hypertrophy. The authors observed opposing expression profiles for ADAMTS10 and ADAMTS17 (Fig. 3F), which is not supported by the results of the okadaic acid experiment (Fig. 3H). Additionally, the rationale behind using 10T1/2 cells and primary chondrocytes from ribs in Fig. 3 is unclear.

#7) In Fig. 4, it is unclear how ATAC-seq data from human tissues helps elucidate the pathological mechanisms in ADAMTS10:ADAMTS17 dKO mice. Instead, performing ATAC-seq using tissues from ADAMTS10:ADAMTS17 dKO mice could reveal genome-wide chromatin accessibility associated with altered chondrogenesis under ADAMTS10 and ADAMTS17 deficiencies.

#8) In Fig. 5, as mentioned in comment #1, it is essential to investigate skin morphology at multiple time points through histological analysis and immunohistochemistry (e.g., KRT1, KRT14, and Loricrin). Additionally, the quantification of fluorescence intensity for Fibrillin-1 in the dKO mice does not align with the images (showing more Fibrillin-1 staining in the

upper panel). Furthermore, it is unclear whether the authors quantified the images with an equal number of cells among the different genotypes (wild-type, ADAMTS10 KO, ADAMTS17 KO, and ADAMTS10:ADAMTS17 dKO). Based on DAPI staining, a significant reduction in cell numbers in dKO mice could contribute to the observed reduction in FN and Fibrillin-1.

#9) In Fig. 7, since COL6 is identified as a potential interacting protein for ADAMTS17, it would be critical to examine whether COL6 expression is decreased in bone and skin tissues from ADAMTS17 KO and/or ADAMTS10:ADAMTS17 dKO mice.

Additional Suggestion

#10) The authors might consider separating this paper into two studies, one focused on bone phenotypes and another on skin phenotypes, to clarify the distinct mechanisms underlying ADAMTS10 and ADAMTS17 deficiencies.

Reviewer #3 (Comments to the Authors (Required)):

In this work, Taye et al. report on the generation and characterization of double knockout (DKO) mice for the Weill-Marchesani syndrome causative genes, Adamts10 and Adamts17. The major finding of this study is that these mice exhibit more severe bone and skin phenotypes compared to the single knockouts. Additionally, the authors investigated the transcriptional regulation of ADAMTS10 and ADAMTS17 in cartilaginous human appendicular skeletal elements and in differentiating 10T1/2 murine chondrocytes. Finally, they propose fibronectin and collagen VI as potential interacting partners and/or substrates for ADAMTS17.

While the rationale for studying the double knockout is clear, the study remains largely descriptive, with many aspects of the data appearing disconnected from each other.

Major Comments:

Stronger evidence is required to confirm the absence of ADAMTS17 protein in the newly generated mouse line. The antibody staining presented in Figure 1C is not convincing. An RNA expression profile and possibly Western blot analysis from tissue or cell lysates are necessary.

Given the exacerbated phenotypes observed in the double knockout mice, which suggest partial compensation in the single knockouts, it would be valuable to investigate whether Adamts10 expression is upregulated in Adamts17 knockout mice.

Figure 3 claims that ADAMTS10 and ADAMTS17 regulate growth plate function and chondrocyte hypertrophy. However, no data is presented to substantiate this statement. The only relevant data, shown in panels A-C, demonstrate reduced growth plate thickness, a phenotype already reported in Adamts17 single knockout mice (Oichi et al., 2019). Because this is a central part of the study, additional data should be provided to support a direct and complementary role of the two proteins in bone development. For example, can the Authors stain for ECM components like fibronectin, fibrillins, col6 in the growth plates of DKO mice? BMP signaling was found to be reduced in Adamts17-deficient growth plates (Oichi et al., 2019). What is the status of BMP signaling in DKO mice?

The Adamts17 staining in Figure 3D should also be performed on knockout samples to verify the specificity of the antibody used.

The experiments in Figure 3K-O were conducted only on single knockout chondrocytes. Since reduced terminal differentiation of Adamts17-deficient chondrocytes has already been reported (Oichi et al., 2019), why were these experiments not extended to the double knockout?

The connection between the expression profiles of Adamts10 and Adamts17 in differentiating pellet cultures of 10T1/2 cells and the in vivo mouse phenotype, as well as the transcriptional regulation in human cartilaginous elements (described in the next section and Figure 4), is unclear. These sections of the manuscript appear unnecessarily complicated and could be streamlined.

In Figure 5A, the micrograph of DKO skin suggests that hair follicles are at different developmental stages, which could significantly influence skin thickness. The authors should clarify this point.

In Figure 5L, the authors should include staining on Adamts17-deficient skin sections to confirm antibody specificity.

In Figure 5M, the immunofluorescence stainings are unconvincing, as the DKO sample appears to contain fewer cells, potentially invalidating the fluorescence intensity quantification. It is unclear whether the reduction in fibrillin-1 and fibronectin fibrils is due to decreased expression, secretion, or assembly. Additional RNA and Western blot analyses should complement the immunostainings to provide a more conclusive result.

Figures 6 and 7: It is difficult to understand how this part relates to the rest of the study, which, according to its title, should focus on Adamts10 and Adamts17 combined inactivation.

Figure 6: Can the authors perform fibronectin Western blot analysis in wild-type and ADAMTS17-deficient fibroblasts to validate their results in a more physiological system?

Figure 7: Authors should stress in the text that they used an antibody directed against the alpha 1 chain of collagen VI, while they identified the alpha 2 as a potential substrate. Is COL6 RNA expression in WMS fibroblasts comparable to the control?

Dear Editor,

We thank the reviewers for the thoughtful critiques and their overall appreciation for the potential importance of the work for the ECM field. As discussed previously, we were unable to perform additional experiments due to lack of wet lab access and funding. However, we modified the figures and text to address most of the reviewer's concerns and believe that the revised manuscript will still contribute robust and important information on the potential roles of ADAMTS10 and ADAMTS17 in the growth plate and skin and during ECM formation. We appreciate the editor's and the reviewer's willingness to work with the authors during these challenging times. Please find below the point-by-point rebuttal of the specific critiques from the reviewers.

Reviewer #1:

Weill-Marchesani is a rare heritable connective tissue disorder, characterized by short stature, brachydactyly, stiff skin and heart and eye anomalies. Autosomal recessive and autosomal dominant forms have been reported. AR can be caused by bi-allelic variants in either ADAMTS10 or ADAMTS17 (among other genes). In order to gain insights into the potentially overlapping/cooperating pathways into which ADAMTS10 and ADAMTS17 function, the authors generated Adamts10/Adamts17 double knockout (DKO) mice, and studied the growth plate and bone formation as well as ECM organization and skin phenotype of these mice, in comparison with WT, single Adamts10, and single Adamts17 KO mice. The combined inactivation of Adamts10 and Adamts17 led to significant postnatal lethality compared to single Adamts10 or Adamts17 KO mice. Strikingly Adamts17 gene dosage had a stronger effect and compromised postnatal survival more than Adamts10 gene dosage.

Question: It is not clear however why these mice die. Could the authors speculate on this?

We speculate that lethality is driven by heart or lung failure. In WMS patients, heart valve dysplasia is observed, in particular in WMS patients harboring *ADAMTS10* mutations. It could be that ADAMTS17 in WMS patients partially compensates for the loss of ADAMTS10 and thus patients maintain sufficient heart valve function. In the absence of both ADAMTS10 and ADAMTS17, heart valve function may be severely compromised in a way that is incompatible with survival. Lung failure could also be a reason that these mice die during early postnatal development and growth, given strong ADAMTS17 and ADAMTS10 mRNA expression that we have reported previously (*Hubmacher et al 2017, Wang et al 2019*). We included a section to discuss these possibilities as part of the discussion.

The combined inactivation of Adamts10 and Adamts17 also exacerbated bone shortening when compared to individual KO mice. The severe bone shortening observed in the DKO mice correlated with a narrower hypertrophic zone in the growth plates.

We agree with this interpretation and consider this one of the key findings of the manuscript.

The authors noted fragile skin during routine handling the DKO mice and observed reduced skin thickness in the single and DKO mice (age between 4-8 weeks), with an exacerbated effect in the DKO, which was predominantly driven by a reduction in the hypodermal layer. These findings indeed suggest that ADAMTS10 and ADAMTS17 are required for skin development, which may relate to abnormal skin

phenotype described in WMS patients. It is however unclear why the DKO show a fragile, thin skin, which contrasts to the stiff and thickened skin described in WMS patients. Increased skin thickness had already been reported previously in ADAMTS10 and ADAMTS17 single KO, but at later age (3-8 months).

Question: Has an age-related change in skin phenotype (changing from thin, fragile to thick, inelastic skin) been described in human WMS patients?

Based on our literature review, thick skin is present in 50 – 74% of WMS patients (Faivre, 2003). These patients had a mean age range from 21.5 – 28 years. An 86-year-old patient was also reported having thick skin (Kutz, 2008). Therefore, there is no evidence in the literature that would suggest thin or fragile skin in WMS patients or age-related changes. The skin phenotype may be rather progressing with age. This is now mentioned in the discussion section.

Comment: I would recommend analyzing the skin in the single and double KO mice at a later time point (e.g. 3 months) to document any changing phenotype.

Since the DKO mice did not survive beyond 3 months of age, we were unable to collect skin samples of skeletally mature or older DKO mice. However, this would be an intriguing question that could be addressed with future conditional ADAMTS10 and ADAMTS17 alleles. For the individual KO mice, we were unable to collect additional samples at later time points due to the corresponding author's lab closure.

N-terminomics and yeast two-hybrid screening assays were used to find substrates for ADAMTS17, and the ECM proteins fibronectin and collagen VI were identified as potential substrates, pointing towards a potentially distinct function for ADAMTS17 as a regulator of ECM formation and homeostasis. However, validation experiments did not reveal direct proteolysis of either fibronectin or collagen VI by ADAMTS17. ECM formation was investigated in primary ADAMTS10- and ADAMTS17-deficient skin fibroblasts and this showed abnormal fibronectin deposition, together with aberrant intracellular accumulation of fibrillin-1. These findings indeed support a role for ADAMTS17 in ECM protein secretion and assembly.

This interpretation is correct. We believe that these more mechanistic findings point towards a possible anabolic role for ADAMTS17 in the formation of the ECM, rather than a more catabolic role that would result in ECM degradation.

Comment: The Identification of fibronectin and Collagen VI as ADAMTS17 binding partners are only briefly discussed in the Discussion Section of the manuscript. This is in imbalance with the other parts. I would recommend expanding this discussion, putting the latter findings (especially about collagen VI) in a broader context.

Based on this suggestion, we expanded the Discussion Section and included a separate paragraph discussing the potential implications of ADAMTS17 regulating the secretion and/or assembly of a key subset of ECM proteins.

Question: The finding of a strong reduction of collagen VI ECM deposition in WMS dermal fibroblasts and concurrent intracellular Collagen VI accumulation is very interesting. Is there any evidence that this (or other) WMS patients suffer from a phenotype that overlaps with Ullrich Congenital Muscular Dystrophy, caused by defects in one of the COL6 genes?

WMS and other acromelic dysplasia feature a (pseudo)muscular built, which is the opposite of the progressive muscle loss observed in COL6-related myopathies and muscular dystrophies. We mention this in the Discussion Section but cannot offer a satisfactory explanation for this apparent discrepancy. We speculate that either the consequences of COL6 mutations are different from dysregulated COL6 deposition in the ECM due to ADAMTS17 mutations, or that the muscle phenotype in WMS is driven by the dysregulation of other ECM proteins. A thorough analysis of skeletal muscle architecture and cell behavior would be required to answer this intriguing question.

This study provides important new insights into the role of ADAMTS10 and ADMATS17 in skin and bone development and homeostasis. Each of the different sections is well-documented with figures and data that strongly support the conclusions. The manuscript is clearly structured and well-written.

We thank the reviewer for the overall positive and encouraging comments.

Minor remarks:

Abstract and introduction:

Conditions (and not variants) are autosomal recessive or autosomal dominant. Variants (or mutations) are bi-allelic or mono-allelic and lead to respectively AR or AD conditions. Please correct this throughout the manuscript

We thank the reviewer for this clarification and corrected the expression throughout the manuscript accordingly.

Page 4 line 66: Homozygosity of a ...variants  variant

This was corrected in the revised manuscript.

Page 5 line 94: While Adamts10 knockout (KO) mice did not result in short stature  ...did not display short stature

This was corrected in the revised manuscript.

Page 5, line 104: The findings of our studies, taken together with prior work strongly suggest  The findings of our studies, taken together with prior work, strongly suggest

A comma was added in the revised manuscript.

Reviewer #2:

In this study, the authors investigate the role of the secreted ADAMTS proteases, ADAMTS10 and ADAMTS17, using double knockout (dKO) mice. The genetic basis of Weill-Marchesani syndrome (WMS) suggests that altered extracellular proteins contribute to its pathogenesis. Based on this, the authors hypothesize that genetic interactions between ADAMTS10 and ADAMTS17 may be linked to WMS phenotypes, including short stature, short hands, and skin anomalies. As the molecular and cellular

mechanisms behind WMS remain unclear, this study provides valuable insights into how these ADAMTS proteases regulate chondrocyte hypertrophy and skin development. Consequently, this work is significant for advancing the field of extracellular matrix biology.

We thank the reviewer for appreciating the potential significance of our manuscript for the ECM field.

Specific Comments

#1) One weakness of this paper is that the authors only examine bone and skin phenotypes at a single time point (4 weeks of age). Around this age, the survival ratio of ADAMTS10:ADAMTS17 dKO mice is approximately 20% (Fig. 1G). As a result, bone phenotypes (Fig. 2A-N), growth plate analysis (Fig. 3A-C), and skin anomalies (Fig. 5A-H) may be indirectly associated with ADAMTS10 and ADAMTS17 deletion. To clarify this, it would be essential to examine these phenotypes at multiple time points, particularly postnatal day 10, when the survivability of dKO mice is affected. Additionally, the use of established markers (e.g., Col2, ColX immunohistochemistry), cell proliferation assays (e.g., Ki-67 staining, BrdU assay), and cell death assays (e.g., Caspase 3 staining, TUNEL assay) would be valuable to examine the chondrogenesis in ADAMTS10:ADAMTS17 dKO mice.

We agree with the reviewer that this is a weakness of the manuscript. Since we do not have access to a live ADAMTS10 or ADAMTS17 colony or lab space, we are unable to analyze earlier timepoints or perform additional immuno-staining. While it is possible that the observed phenotypic changes are indirectly related to ADAMTS10 and/or ADAMTS17 deletion, the examination of earlier time points would not resolve this point. In fact, identifying the direct consequences of ADAMTS10 or ADAMTS17 deletion is the most challenging aspect of this work and other published studies. Under the assumption, that ADAMTS10 and ADAMTS17 are proteases, their biology is defined by the biology of their substrates. Therefore, we included the documentation of our efforts to identify potential substrates for ADAMTS17 as part of this manuscript.

#2) In Fig. 1C, the authors should use ADAMTS17 KO tissue (rather than dKO) in growth plate and skin fibroblast analysis to examine the reduction of ADAMTS17 expression. The deletion of ADAMTS10 could potentially suppress ADAMTS17 expression indirectly.

In the skin, the ADAMTS17 signal was strongest around hair follicles, which is consistent with the in-situ hybridization data shown in Fig. 5. This signal was absent in the ADAMTS17 KO tissue, which is strong evidence for the specificity of the ADAMTS17 antibody and the successful deletion/ablation of ADAMTS17. Since ADAMTS10 is also expressed in skin (Wang et al 2019), this data set demonstrates in addition that the ADAMTS17 antibody does not cross-react with ADAMTS10. We agree that the deletion of ADAMTS10 could indeed suppress ADAMTS17 indirectly. However, this would only be relevant if we would use the ADAMTS17 antibody on ADAMTS10 KO tissue and do not obtain a signal. Therefore, strong reduction of the ADAMTS17 signal around dKO hypertrophic chondrocytes and absence in dKO skin fibroblasts lend additional support to the specificity of the ADAMTS17 antibody used here for immunohistochemistry.

#3) In Fig. 3A, presenting low-magnification H&E images would be beneficial. While the authors attribute the bone phenotype to growth plate retardation, ADAMTS17 may also regulate osteogenesis and osteoclastogenesis. To support this, dKO mice show less trabecular bone and/or bone marrow compared with wild-type mice (Fig. 3A, right panel).

We included low magnification images in the revised version as new Fig. 3A. We also indicate that the changes in the bone marrow are attributed to preparation artifacts likely introduced during tissue preparation, in particular paraffin embedding and sectioning. This is based on the inconsistency of this feature when looking at the additional images from WT and DKO growth plates. We agree with the reviewer that it would be worthwhile analyzing osteogenesis and osteoclastogenesis in the DKO compared to WT mice, which in fact was part of a recent, unfunded NIH grant application. While osteoporosis is not recognized as a feature of Weill-Marchesani syndrome, there is one case report suggesting primary osteoporosis in a patient with WMS (Giordano 1997, PMID: 9075633).

#4) To strengthen the authors' interpretation, Safranin O staining should be performed to visualize the morphology of the growth plate.

Since we did not perform Safranin O staining, we included images from Masson trichrome-stained growth plate sections (new Fig. 3B). These images confirm overall growth plate disorganization and suggest no apparent changes in the collagen content of the growth plate. However, collagen content seemed to be more intense in the underlying bone sections, which would warrant further investigation.

#5) The ADAMTS17 immunostaining should be improved with appropriate negative controls, as there is a discrepancy between the results shown in Fig. 1C (growth plate in wild-type) and Fig. 3D.

We respectfully disagree with this comment. Negative controls are shown in Fig. 1C, where the signal originating from the ADAMTS17 antibody was strongly reduced in the corresponding KO and dKO tissues. Figures 1C and 3D both show intense staining for ADAMTS17 around hypertrophic chondrocytes in WT tissue. Staining in the mineralized tissue underneath the hypertrophic zone is considered non-specific. The intention of Fig. 1C is to show the specificity of the ADAMTS17 antibody, while in Fig. 3D, we highlight ADAMTS17 localization in the pericellular matrix of hypertrophic chondrocytes compared to proliferating/columnar chondrocytes in the normal, i.e. WT growth plate. In the revised manuscript, we cross-reference Fig. 1C in the Results Section for Fig. 3D to enhance clarity.

#6) There is a logical disconnect in the use of okadaic acid treatment in primary chondrocytes from ribs to explore the molecular link between ADAMTS10 and ADAMTS17 in chondrocyte hypertrophy. The authors observed opposing expression profiles for ADAMTS10 and ADAMTS17 (Fig. 3F), which is not supported by the results of the okadaic acid experiment (Fig. 3H). Additionally, the rationale behind using 10T1/2 cells and primary chondrocytes from ribs in Fig. 3 is unclear.

We acknowledge that the 10T1/2 cells are inferior to primary chondrocytes and removed data obtained with the 10T1/2 cells from the revised manuscript.

#7) In Fig. 4, it is unclear how ATAC-seq data from human tissues helps elucidate the pathological mechanisms in ADAMTS10:ADAMTS17 dKO mice. Instead, performing ATAC-seq using tissues from

ADAMTS10:ADAMTS17 dKO mice could reveal genome-wide chromatin accessibility associated with altered chondrogenesis under ADAMTS10 and ADAMTS17 deficiencies.

We agree with the reviewer that this data set does not help explain the pathogenesis of WMS. However, it may contribute to a better understanding of ADAMTS10 and ADAMTS17 regulation and function. Our intent is to show that ADAMTS10 and ADAMTS17 are likely regulated during human growth plate development on a transcriptional level. We will include additional text to enhance the understanding of the sites in the skeleton, which paint a picture of potential functional roles for ADAMTS10 and ADAMTS17 in humans. We believe that performing ATAC-seq on DKO and WT tissues is less relevant considering that ADAMTS10 and ADAMTS17 are not transcription factors or chromatin modifiers. Therefore, any effects would be indirect rather than direct and challenging to interpret.

#8) In Fig. 5, as mentioned in comment #1, it is essential to investigate skin morphology at multiple time points through histological analysis and immunohistochemistry (e.g., KRT1, KRT14, and Loricrin). Additionally, the quantification of fluorescence intensity for Fibrillin-1 in the dKO mice does not align with the images (showing more Fibrillin-1 staining in the upper panel). Furthermore, it is unclear whether the authors quantified the images with an equal number of cells among the different genotypes (wild-type, ADAMTS10 KO, ADAMTS17 KO, and ADAMTS10:ADAMTS17 dKO). Based on DAPI staining, a significant reduction in cell numbers in dKO mice could contribute to the observed reduction in FN and Fibrillin-1.

We are unable to perform additional immunostaining based on lack of access to a wet lab. Additional time points are also unavailable. Regarding the immunofluorescence quantification, in our experience, DAPI intensity correlates very tightly with cell numbers. Therefore, normalization of the immunofluorescence signal to cell number rather than DAPI intensity will result in the same outcome. However, we now acknowledge that the reduction in fibrillin-1 and fibronectin ECM deposition could be attributed to reduced cell numbers. In addition, we included a section about the limitations of our studies in the discussion section. There, we acknowledge that without measuring gene expression and protein secretion, we are unable to pinpoint the causes of reduced ECM deposition in the KOs and the DKO fibroblasts. Unfortunately, we are unable to perform such experiments.

#9) In Fig. 7, since COL6 is identified as a potential interacting protein for ADAMTS17, it would be critical to examine whether COL6 expression is decreased in bone and skin tissues from ADAMTS17 KO and/or ADAMTS10:ADAMTS17 dKO mice.

We stained skin sections from 4-month-old WT, *Adamts17* KO and DKO mice with an antibody against Col VI (see below). We observed Col VI localization around the shaft or hair follicles in the WT with little staining in the interstitial ECM. This would be consistent with the potential localization of ADAMTS17 as suggested by in-situ hybridization and immunostaining. In the DKO skin, we observed a similar localization around hair follicles. However, the staining appeared more diffuse around the hair follicles, and we observed additional Col VI immunoreactivity in the spaces between the follicles. Due to the small number of DKO animals analyzed and the various orientations of the hair follicles and hair shafts, we are not comfortable to include these data in the manuscript.

Additional Suggestion

#10) The authors might consider separating this paper into two studies, one focused on bone phenotypes and another on skin phenotypes, to clarify the distinct mechanisms underlying ADAMTS10 and ADAMTS17 deficiencies.

We acknowledge that mechanisms driving the distinct bone and skin phenotypes may be tissue specific. Nevertheless, we believe that the value of the manuscript is to show a genetic interaction between ADAMTS10 and ADAMTS17 in vivo and to point towards potential substrates. In addition, WMS is a syndromic disorder and therefore reporting the skin and bone phenotypes makes the manuscript more complete.

Reviewer #3:

In this work, Taye et al. report on the generation and characterization of double knockout (DKO) mice for the Weill-Marchesani syndrome causative genes, *Adamts10* and *Adamts17*. The major finding of this study is that these mice exhibit more severe bone and skin phenotypes compared to the single knockouts. Additionally, the authors investigated the transcriptional regulation of ADAMTS10 and ADAMTS17 in cartilaginous human appendicular skeletal elements and in differentiating 10T1/2 murine chondrocytes. Finally, they propose fibronectin and collagen VI as potential interacting partners and/or substrates for ADAMTS17. While the rationale for studying the double knockout is clear, the study remains largely descriptive, with many aspects of the data appearing disconnected from each other.

We acknowledge the largely descriptive nature of the study. Nevertheless, we believe that we report important information for the extracellular matrix and rare disease community. In addition, we attempted to not only identify potential substrates by mass spectrometry but also validate candidate substrates through biochemical and cell culture experiments. We think that the finding that fibronectin and collagen VI may not be direct substrates for ADAMTS17 are important and point towards a more complex role for ADAMTS17 in regulating extracellular matrix formation.

Major Comments:

Stronger evidence is required to confirm the absence of ADAMTS17 protein in the newly generated mouse line. The antibody staining presented in Figure 1C is not convincing. An RNA expression profile and possibly Western blot analysis from tissue or cell lysates are necessary.

We include qPCR data as new Fig. 1C to show significant reduction of ADAMTS17 mRNA in *Adamts17* KO tissue. Together with the reduction of ADAMTS17 protein in tissues (Fig. 1D), we believe that the combined evidence supports the successful generation of an *Adamts17* knockout model. The text and figure legend were updated accordingly.

Given the exacerbated phenotypes observed in the double knockout mice, which suggest partial compensation in the single knockouts, it would be valuable to investigate whether *Adamts10* expression is upregulated in *Adamts17* knockout mice.

This is an interesting question, which we did not address in the manuscript and are unable to do so for the revision due to the lack of live mouse colonies. We would potentially predict that *Adamts17* is upregulated in *Adamts10* KO mice since very mild bone length phenotypes of *Adamts10* KO mice seemed to be exacerbated by the additional lack of one allele of *Adamts17*. However, even if there is no compensatory upregulation, the presence of at least one of the combined four alleles seems to alleviate most phenotypes, including bone length and postnatal survival.

Figure 3 claims that ADAMTS10 and ADAMTS17 regulate growth plate function and chondrocyte hypertrophy. However, no data is presented to substantiate this statement. The only relevant data, shown in panels A-C, demonstrate reduced growth plate thickness, a phenotype already reported in *Adamts17* single knockout mice (Oichi et al., 2019). Because this is a central part of the study, additional data should be provided to support a direct and complementary role of the two proteins in bone development. For example, can the Authors stain for ECM components like fibronectin, fibrillins, col6 in the growth plates of DKO mice? BMP signaling was found to be reduced in *Adamts17*-deficient growth plates (Oichi et al., 2019). What is the status of BMP signaling in DKO mice?

In response to the critique, we toned down the statements and used more descriptive language to report these data. We believe that the difference in growth plate architecture and the differential chondrocyte behavior are interesting observations worth reporting.

The *Adamts17* staining in Figure 3D should also be performed on knockout samples to verify the specificity of the antibody used.

These data are part of Fig. 1D, where we show absence of ADAMTS17 staining in the ADAMTS17 KO and in dKO. If the reviewer would prefer moving this panel from Fig. 1D into Fig. 3D we can do this. However, we prefer to present Fig. 1D early in the manuscript to provide evidence for the absence of ADAMTS17 in the KO and thus as validation of the *Adamts17* KO model.

The experiments in Figure 3K-O were conducted only on single knockout chondrocytes. Since reduced terminal differentiation of *Adamts17*-deficient chondrocytes has already been reported (Oichi et al., 2019), why were these experiments not extended to the double knockout?

Since we observed very little terminal differentiation in the single *Adamts17* KO, we had no reason to believe that this would be different in the DKO. DKO mice are challenging to get, and we would have needed to isolate chondrocytes prior to genotyping. We also do not have a live mouse colony anymore.

The connection between the expression profiles of *Adamts10* and *Adamts17* in differentiating pellet cultures of 10T1/2 cells and the in vivo mouse phenotype, as well as the transcriptional regulation in human cartilaginous elements (described in the next section and Figure 4), is unclear. These sections of the manuscript appears unnecessarily complicated and could be streamlined.

Following this critique, we streamlined these sections and made the connections clearer. In addition, we removed the data obtained with 10T1/2 cells in the revised manuscript. Our intent is to show that ADAMTS10 and ADAMTS17 are likely regulated during human growth plate development on a transcriptional level.

In Figure 5A, the micrograph of DKO skin suggests that hair follicles are at different developmental stages, which could significantly influence skin thickness. The authors should clarify this point.

The skin samples were all collected at the same age (4 weeks old). The different appearance of hair follicles may be related to the plane of sectioning. However, this would not affect the thickness of the skin or its individual layers.

In Figure 5L, the authors should include staining on *Adamts17*-deficient skin sections to confirm antibody specificity.

Antibody-specificity was determined in other tissues and with skin fibroblasts (Fig. 1D). In addition, the signal correlates with the mRNA expression pattern. Therefore, we believe that the strong signal around hair follicles is originating from ADAMTS17.

In Figure 5M, the immunofluorescence staining is unconvincing, as the DKO sample appears to contain fewer cells, potentially invalidating the fluorescence intensity quantification. It is unclear whether the reduction in fibrillin-1 and fibronectin fibrils is due to decreased expression, secretion, or assembly. Additional RNA and Western blot analyses should complement the immunostainings to provide a more conclusive result.

In our experience, DAPI intensity correlates very tightly with cell numbers. Therefore, normalization of the immunofluorescence signal to cell number rather than DAPI intensity will result in the same outcome. However, we now mention that the reduction in fibrillin-1 and fibronectin ECM deposition could be attributed to reduced cell numbers. In addition, we included a section about the limitations of our studies at the end of the discussion section. We acknowledge that without measuring gene expression and protein secretion, we are unable to pinpoint the exact causes of reduced ECM deposition in the KOs and the DKO fibroblasts. Unfortunately, we are unable to perform these experiments.

Figures 6 and 7: It is difficult to understand how this part relates to the rest of the study, which, according to its title, should focus on *Adamts10* and *Adamts17* combined inactivation.

Since ADAMTS10 and ADAMTS17 are considered to function as proteases, it is important to attempt to identify individual substrates. Therefore, we believe that both figures provide

important information and contribute to the impact of the manuscript for the ECM and protease community.

Figure 6: Can the authors perform fibronectin Western blot analysis in wild-type and ADAMTS17-deficient fibroblasts to validate their results in a more physiological system?

Unfortunately, we are unable to perform additional cell culture experiments or tissue extractions due to lack of access to laboratory space and reagents and the shutdown of the live mouse colony.

Figure 7: Authors should stress in the text that they used an antibody directed against the alpha 1 chain of collagen VI, while they identified the alpha 2 as a potential substrate. Is COL6 RNA expression in WMS fibroblasts comparable to the control?

This information is now specified in the figure legend related to Fig. 7 and in the description of the results in the Result Section.

September 11, 2025

RE: Life Science Alliance Manuscript #LSA-2025-03232R

Dr. Dirk Hubmacher
Icahn School of Medicine at Mount Sinai
Orthopedics
1 Gustave L. Levy Pl
New York 10029

Dear Dr. Hubmacher,

Thank you for submitting your revised manuscript entitled "Combined ADAMTS10 and ADAMTS17 inactivation exacerbates bone shortening and skin phenotypes". Your revised manuscript was evaluated by two of the original reviewers whose comments are appended below. As you will note, both reviewers acknowledge the toned down conclusions in your revised version.

We would be happy to publish your paper in Life Science Alliance pending final revisions necessary to meet our formatting guidelines. Along with points mentioned below, please tend to the following:

- All claims must be supported with data, so please remove the phrase "data not shown" in line 359. Either include these data, or remove the related claim.
- In the methods section, please provide details for:
 - a. Masson's trichrome stain method (source, concentration of stain or citation for the method).
 - b. Microscopy (name of microscope used, nature of microscopy, objective used, NA etc).
 - c. anti-myc antibody (name of antibody and source).
 - d. analyses in Fiji/ImageJ (name of plug-in used, citation).
- Please specify in the 'Data Availability' section if data for mass spectrometry (MS)-based N-terminomics approach is uploaded to any repository or is made available in supplemental section.
- Please upload a clean manuscript file without track changes.
- It is recommended to exclude figures from the manuscript text and upload them separately.
- Please add your main and table legends to the main manuscript text after the references section.
- We encourage you to revise the figure legends for Figure 6 such that the figure panels are introduced in alphabetical order.
- In the text of the manuscript, a reference should be cited by author and year of publication; 'et al' should be used if there are more than two authors (i.e., Smith & Jones, 2003; Smith et al, 2000). Alternatively, in the text, a reference can be cited by a number in parentheses: (1). The citation should appear within the punctuation. Where more than one reference is cited, commas should be used to separate references, e.g.: (1, 2, 3, 8).
- In the reference list, citations should be listed with the authors' surnames and initials inverted. Where there are more than 10 authors on a paper, the first 10 will be listed, followed by 'et al.'
- Please add the X and Bluesky handles of your host institute/organisation, as well as your own and/or one of the authors in our system.
- Please be sure that the authorship listing and order is correct.

LSA now encourages authors to provide a 30-60 second video where the study is briefly explained. We will use these videos on social media to promote the published paper and the presenting author (for examples, see <https://docs.google.com/document/d/1-UWCfbE4pGcDdcgzcmiuJI2XMBJnxKYeqRvLLrLS08s/edit?usp=sharing>). Corresponding or first-authors are welcome to submit the video. Please submit only one video per manuscript. The video can be emailed to contact@life-science-alliance.org

A. FINAL FILES:

B. MANUSCRIPT ORGANIZATION AND FORMATTING:

Thank you for your attention to these final processing requirements. Please revise and format the manuscript and upload materials as soon as you are able.

Sincerely,

Sarita Hebbar, PhD
Scientific Editor
Life Science Alliance
<http://www.lsjournal.org>

Reviewer #2 (Comments to the Authors (Required)):

While the authors were unable to fully address the questions raised by this reviewer due to limited material resources, I believe this manuscript is still worth publishing to enhance the understanding of the roles of ADAMTS10 and ADAMTS17 in bone and skin development.

Reviewer #3 (Comments to the Authors (Required)):

In their revised version of the manuscript, the authors have made a genuine effort to address the majority of my previous concerns. Although they were unable to perform additional experiments due to the closure of their laboratory, they have provided thoughtful clarifications and strengthened the discussion where possible. While a few questions remain unresolved, I believe that the manuscript nonetheless offers a substantial body of novel data and reflects significant effort. On balance, I consider it a valuable contribution to the field and recommend it for publication.

Dear Editor,

We thank the editor and reviewers for their generous support of this manuscript. Please find below the comments on the requested changes.

All claims must be supported with data, so please remove the phrase "data not shown" in line 359. Either include these data or remove the related claim.

The statement has been deleted.

In the methods section, please provide details for:

- Masson's trichrome stain method (source, concentration of stain or citation for the method).

We included the kit that we used.

- Microscopy (name of microscope used, nature of microscopy, objective used, NA etc).

This information was added.

- anti-myc antibody (name of antibody and source).

Name and product number have been added.

- analyses in Fiji/ImageJ (name of plug-in used, citation).

We did not use a particular plug-in. The reference for Fiji/ImageJ is included at the first mention in the Materials & Methods section (Histology section).

Please specify in the 'Data Availability' section if data for mass spectrometry (MS)-based N-terminomics approach is uploaded to any repository or is made available in supplemental section.

The mass spectrometry proteomics data have been deposited to the ProteomeXchange Consortium via the PRIDE partner repository with the dataset identifier PXD068317 and 10.6019/PXD06831767. The data set will be released to the public within 24h after publication of the manuscript.

The data sets are currently available for reviewers/editors by logging in to the PRIDE website using the following details: Project accession: PXD068317, Token: hLO0CW7pXxX8, or logging in to the PRIDE website using the following account details: Username: reviewer_pxd068317@ebi.ac.uk, Password: DsYXn2h1arMv.

Please upload a clean manuscript file without track changes.

Track-changes have been removed in the final uploaded manuscript.

It is recommended to exclude figures from the manuscript text and upload them separately.

Figures were uploaded separately and excluded from the manuscript text.

Please add your main and table legends to the main manuscript text after the references section.

The figure legends were moved to the end of the manuscript after the reference section.

We encourage you to revise the figure legends for Figure 6 such that the figure panels are introduced in alphabetical order.

The figure legend was updated to have the panels in alphabetical order.

In the text of the manuscript, a reference should be cited by author and year of publication; 'et al' should be used if there are more than two authors (i.e., Smith & Jones, 2003; Smith et al, 2000). Alternatively, in the text, a reference can be cited by a number in parentheses: (1). The citation should appear within the punctuation. Where more than one reference is cited, commas should be used to separate references, e.g.: (1, 2, 3, 8).

The references have been updated accordingly using the Life Science Alliance style format from EndNote.

In the reference list, citations should be listed with the authors' surnames and initials inverted. Where there are more than 10 authors on a paper, the first 10 will be listed, followed by 'et al.'.

This change is included by using the appropriate EndNote style file.

Please add the X and Bluesky handles of your host institute/organisation, as well as your own and/or one of the authors in our system.

The information was added.

Please be sure that the authorship listing and order is correct.

The authorship order is correct.

No press release is planned.

LSA now encourages authors to provide a 30-60 second video where the study is briefly explained. We will use these videos on social media to promote the published paper and the presenting author. Corresponding or first-authors are welcome to submit the video. Please submit only one video per manuscript. The video can be emailed to contact@life-science-alliance.org.

Not decided.

FINAL FILES:

An editable version of the final text (.DOC or .DOCX) is needed for copyediting (no PDFs).

A word file was uploaded.

High-resolution figure, supplementary figure and video files uploaded as individual files: See our detailed guidelines for preparing your production-ready images, <https://www.life-science-alliance.org/authors>.

High resolution figures were uploaded in the Tiff format.

Summary blurb (enter in submission system): A short text summarizing in a single sentence the study (max. 200 characters including spaces). This text is used in conjunction with the titles of papers, hence should be informative and complementary to the title. It should describe the context and significance of the findings for a general readership; it should be written in the present tense and refer to the work in the third person. Author names should not be mentioned.

A summary blurb was added to the manuscript file and in the submission system.

MANUSCRIPT ORGANIZATION AND FORMATTING:

For all the western blots and protein gels except for panel 7M, the full lengths of the individual lanes are shown in the main figure.

September 17, 2025

RE: Life Science Alliance Manuscript #LSA-2025-03232RR

Dr. Dirk Hubmacher
Icahn School of Medicine at Mount Sinai
Orthopedics
1 Gustave L. Levy Pl
New York 10029

Dear Dr. Hubmacher,

Thank you for submitting your Research Article entitled "Combined ADAMTS10 and ADAMTS17 inactivation exacerbates bone shortening and skin phenotypes". It is a pleasure to let you know that your manuscript is now accepted for publication in Life Science Alliance. Congratulations on this interesting work.

Your manuscript will now progress through copyediting and proofing. It is journal policy that authors provide original data upon request. Whilst evaluating the proofs, we encourage you to include the details for objectives used for fluorescence microscopy (if different from those provided under "histology" section of materials and methods).

DISTRIBUTION OF MATERIALS:

Again, congratulations on a very nice paper. I hope you found the review process to be constructive and are pleased with how the manuscript was handled editorially. We look forward to future exciting submissions from your lab.

Sincerely,

Sarita Hebbar, PhD
Scientific Editor
Life Science Alliance
<http://www.lsajournal.org>